# Evaluating and Steering Modality Preferences in Multi-modal LLMs

**Yu Zhang** [1 2]   **Jinlong Ma** [1]   **Yongshuai Hou** [2]   **Xuefeng Bai** [1]   **Kehai Chen** [1 2]
**Yang Xiang** [2]   **Jun Yu** [1]   **Min Zhang** [1 2]

## Abstract

Multi-modal large language models (MLLMs) have achieved remarkable success on complex multi-modal tasks. However, it remains insufficiently explored whether they exhibit *modality preference*, a tendency to favor one modality over another when processing multi-modal contexts. To study this question, we introduce **MC²** benchmark, which constructs controlled evidence-conflict scenarios to systematically evaluate modality preference in decision-making. Extensive experiments reveal that all 20 tested MLLMs generally demonstrate clear modality preferences, and such preferences are statistically associated with performances of downstream taks for MLLMs. Further analysis shows that modality preference can be controlled by instruction guidance and captured within the latent representations of MLLMs. Built on these insights, we propose a probing and steering method based on representation engineering to explicitly control modality preference without requiring additional fine-tuning. This method effectively amplifies modality preference toward a desired direction and demonstrates promising improvements across multiple multi-modal understanding and reasoning tasks.[1]

## 1. Introduction

Multi-modal Large Language Models (MLLMs; Achiam et al., 2023; Zhu et al., 2026b; Zhang et al., 2025b; Wei et al., 2026; Zhu et al., 2026a; Wang et al., 2026; Zhang et al., 2025c) have emerged as a powerful paradigm for processing and reasoning across heterogeneous data

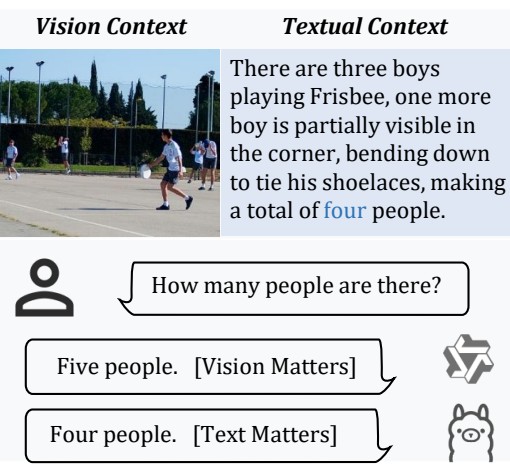

*Figure 1.* Illustrations of evaluating modality preference. Using multi-modal conflict context pairs to evaluate modality preference.

modalities (e.g., text, images, video). Recent advances demonstrate their exceptional capabilities on complex tasks with multi-modal contexts, such as autonomous web browsing (He et al., 2024), graphical user interface understanding (Hong et al., 2024b), and multi-modal dialogue systems (Sun et al., 2022; Lin et al., 2026). Despite impressive performance, fundamental questions remain about their *modality preference*—whether MLLMs tend to rely more heavily on one modality than others, and to what extent they favor a specific modality when resolving multi-modal inputs.

To investigate this, one line of work (Fu et al., 2024; Amara et al., 2024) compares model performance on unimodal input, providing either only text or only image input for the same question. Another line of research analyzes the relative contributions of textual and visual context, typically by removing one modality to observe the changes in the downstream performance (Park et al., 2025) or Shapley value (Alishahi et al., 2019; Parcalabescu & Frank, 2024; 2022). However, such settings inherently introduce bias, as *they isolate modalities, thus failing to reflect how models process inputs in realistic multi-modal scenarios*, where information from different modalities naturally co-occurs.

In this paper, we provide a controllable setup to study the

---

[1]Harbin Institute of Technology, Shenzhen, China [2]Peng Cheng Laboratory, Shenzhen, China. Correspondence to: Xuefeng Bai <baixuefeng@hit.edu.cn>.

*Proceedings of the 43rd International Conference on Machine Learning*, Seoul, South Korea. PMLR 306, 2026. Copyright 2026 by the author(s).

[1]The code and data are released at https://github.com/EchoDreamer/Modality-Preference.

modality preference in MLLMs. As shown in the Figure 1, we introduce a modality context conflict setting, where MLLMs are asked to answer a question based on a pair of contrasting pieces of evidence from different modalities. In this way, we can determine modality preference based on the answer given by MLLMs. To enable a rigorous and fair assessment, we eliminate existing confounding factors including the internal knowledge of MLLMs and inherent understanding capabilities. Consequently, we curate and annotate the dataset to ensure that the model achieves precise question and single-modality context comprehension. Building upon this, we introduce a semi-automated annotation framework to construct a refined **M**odality **C**ontext **C**onflict dataset, **MC²**, which covers eight tasks with 2,000 carefully selected samples. Using $MC^2$, we conduct a comprehensive analysis of modality preference across a diverse set of 20 representative MLLMs. Our study reveals several intriguing findings:

1) Most MLLMs (except Qwen2.5-VL and InternVL3) display text preference, as shown in the Figure 2, and modality preference are statistically associated with the performance of downstream tasks.
2) Internal attention patterns give rise to modality preference, and the underlying factors can be traced to the training data recipe and the scale of MLLMs.
3) The expression of modality preference is both steerable through explicit instruction guidance and geometrically separable in the latent space.

Built on these, we propose a modality preference probing and steering method based on representation engineering (Zou et al., 2023) to explicitly amplify the modality preference without additional fine-tuning. Experimental results show that the proposed method leads to notable performance improvements on multi-modal understanding and reasoning tasks. Our main contributions are summarized as follows:

1) We rigorously evaluate modality preference in scenarios where visual and textual evidence co-exist, and explicitly decouple it from confounding factors.
2) We characterize the properties of modality preference, investigate its underlying causes, and derive new insights for modality learning of MLLMs.
3) We propose a training-free method that steers modality preference, enabling controllable preference adjustment and enhancing performance on downstream tasks.

## 2. Related Work

### 2.1. Modality Preference

In recent years, Large Language Models have achieved impressive results across a wide spectrum of tasks (Lu et al., 2022; Zhang et al., 2024b; Guo et al., 2024; Xu et al., 2025; Liang et al., 2026a;b; Yan et al., 2026; Zhao et al., 2025; Zhu et al., 2025). Building upon these, multi-modal Large Language Models have extended such capabilities, demonstrating strong performance in visual question answering (Zhang et al., 2026a; 2025a; Liu & Liu, 2025; Wei et al., 2023; Zhu et al., 2026a; 2024) and visual reasoning (Wang et al., 2025a; Wei et al., 2026a; Huang et al., 2025; Wang et al., 2025b; Xu et al., 2026). Recent research has turned to systematically evaluating their failure modes (Wei et al., 2024a; Amara et al., 2024; Alishahi et al., 2019; Chen et al., 2024b; Zhang et al., 2026b), aiming to better guide the development of more reliable models.

Existing studies on the modality preference of MLLMs can be broadly divided into two categories: 1) investigating the data-related factors that give rise to modality preference or bias, and 2) analyzing the intrinsic characteristics of modality preference within models.

**Data factors influencing modality preference.** Research on data-related factors (Guo et al., 2023; Chen et al., 2024a; Leng et al., 2024) explores how properties of multi-modal datasets give rise to and reinforce modality preference. In particular, single-modality-solvable samples in multi-modal datasets can bias optimization dynamics, encouraging models to over-rely on one modality (Chen et al., 2024a). Furthermore, their inclusion in evaluation benchmarks artificially inflates performance metrics, as models can exploit these unimodal shortcuts instead of performing genuine cross-modal integration (Leng et al., 2024; Ma et al., 2026). While these studies establish data's role in inducing preference, our work focuses on exploring the intrinsic modality preference inherent in MLLMs themselves, independent of specific data distributions.

**Evaluating the intrinsic modality preference in MLLMs.** Early studies (Peng et al., 2022; Yang et al., 2024; Wei et al., 2024b) analyze modality preference or bias by examining how multi-modal models optimize for multi-modal inputs. Through such analyses, researchers observe that modality bias has a significant impact on both model optimization and downstream task performance (Peng et al., 2022; Ren et al., 2022; Zhang et al., 2024a). While these works offer valuable insights, they typically require training models from scratch, which makes them impractical for large-scale multi-modal systems. Recent studies have investigated intrinsic modality preference in MLLMs by evaluating model performance on unimodal inputs—using only text or only image for the same task (Fu et al., 2024; Amara et al., 2024)—and by applying Shapley value-based attribution methods to quantify the contribution of each modality (Alishahi et al., 2019; Parcalabescu & Frank, 2022; 2024). However, in real-world multi-modal applications, all modalities are indispensable for task resolution, making these frameworks

*inadequate for determining true modality preference.* (Wu et al., 2025) evaluate the model's ability to detect conflict under scenarios involving conflicting multi-modal contexts. However, conflict detection is only one facet of multi-modal reasoning and does not comprehensively reflect a model's modality preference when processing multi-modal contexts.

Unlike prior studies, we decouple modality preference from confounding variables—such as internal model knowledge and varying understanding capabilities—by ensuring the models can accurately comprehend the provided context.

## 2.2. Representation Engineering

Extensive research has shown that LLMs encode interpretable concepts, such as sentiment, truthfulness, and stylistic attributes in representation space in LLMs (Pan-ickssery et al., 2023; Subramani et al., 2022; Turner et al., 2023). Building on this foundation, representation engineering has proven effective for editing, enhancing, or suppressing specific behaviors in LLMs (Greenblatt et al., 2023; Stolfo et al., 2024; Wu et al., 2024; Xu et al., 2024; Zou et al., 2023). In this work, we extend this paradigm to a novel setting: controlling modality preference in MLLMs. Instead of focusing on abstract properties, our method identifies and manipulates representation directions that are sensitive to modality preference, allowing for flexible and precise modulation of model behavior.

## 3. The MC² Benchmark

In this section, we introduce the design and methodology behind the construction of the **M**ultimodal **C**ontext **C**onflict dataset, **MC²**, intended for evaluating modality preference. We outline the data design philosophy in Section 3.1, followed by the data construction pipeline in Section 3.2 and the question design and evaluation metric in Section 3.3.

## 3.1. Data Design Philosophy

Modality preference is a fundamental behavioral tendency to favor one modality over another, irrespective of MLLMs's internal knowledge and inherent understanding capabilities. To enable a rigorous and fair assessment, we eliminate these confounding factors by ensuring that models can accurately comprehend the multi-modal context and respond based solely on the provided information. We elaborate on this design choice below: 1) Neutralizing Knowledge Bias: By ensuring models can derive answers directly from the context, we decouple modality preference from the interference of varying internal knowledge bases. 2) Establishing a Level Playing Field: By verifying that all models can comprehend the multi-modal context, we establish a "common ground"—a baseline of competence that ensures a fair comparison across MLLMs with diverse

capabilities. Finally, we construct modality-conflict pairs to evaluate and compare the modality preferences of different MLLMs in a controlled environment.

## 3.2. Semi-automated Data Construction Pipeline

In this section, we introduce our semi-automated data construction pipeline, which follows a meticulous process to ensure the reliability of the dataset. The data is derived from the TDIUC (Kafle & Kanan, 2017), sourced from MS-COCO (Lin et al., 2014). Therefore, the data is widely adopted in model development, ensuring that most MLLMs can recognize the images. Firstly, we select the image as vision context $c^v$, question $q$, and answer $A^v$ based on the image and the caption $cap$ for each sample from TDIUC. Then the pipeline follows these steps:

**Textual Context Construction.** Given a sample including $c^v$, $q$, $A^v$ and $cap$, we construct candidate contrastive textual contexts $c^t$ that conflict with $c^v$ specifically in relation to $q$ but are aligned with $c^v$ and $cap$ in terms of overall scene semantics. We prompt DeepSeekV3 (Liu et al., 2024a) and ChatGPT4o-mini (Hurst et al., 2024b) to generate a distractor answer $A^t$ to $q$, together with $c^t$ that plausibly supports $A^t$, using carefully crafted instructions. For each model, we generate two pairs of $A^t$ and $c^t$ to facilitate downstream data selection. To ensure that all evaluated MLLMs can correctly recognize both visual and textual contexts, we employ several weak MLLMs, such as LLaVA1.5-7B (Liu et al., 2024b) and QwenVL-7B (Bai et al., 2023), as judges to select samples that can be correctly understood with respect to $c^v$ and $c^t$.

**Human Verification.** We incorporate manual inspection to ensure the high quality of the data annotation. Specifically, we verify the existence of conflicts between $c^v$ and $c^t$ and ensure that both contexts can correctly direct $q$ to the corresponding answers, $A^v$ and $A^t$. Each sample is cross-verified by three human annotators to ensure the reliability of the results, and when errors are found, annotators either correct or discard the sample entirely.

**Iterative Refinement.** The dataset undergoes multiple rounds of refinement through a feedback loop between textual context generation and human verification, which rectifies potential errors, thus enhancing the data quality.

**MC².** To this end, we develop **MC²**, a modality-conflict dataset comprising 2,000 samples spanning diverse tasks. The detailed construction procedures for textual context generation, the manual annotation, the data format and dataset statistics are provided in Apdx B.

## 3.3. Question Design and Evaluation Metric

**Question Design.** We reformulate the original questions with ChatGPT-4o-mini (Hurst et al., 2024a) into a binary-

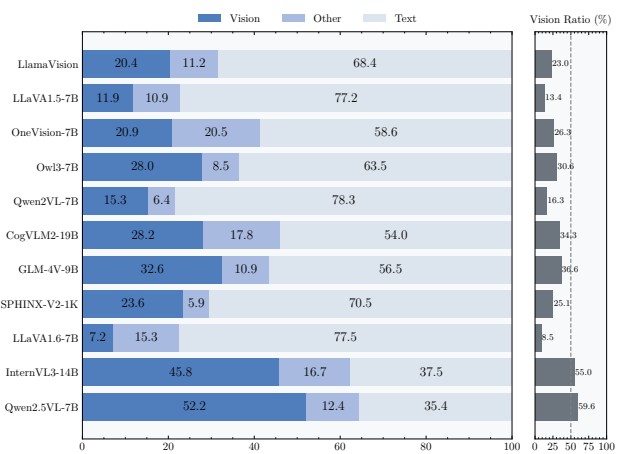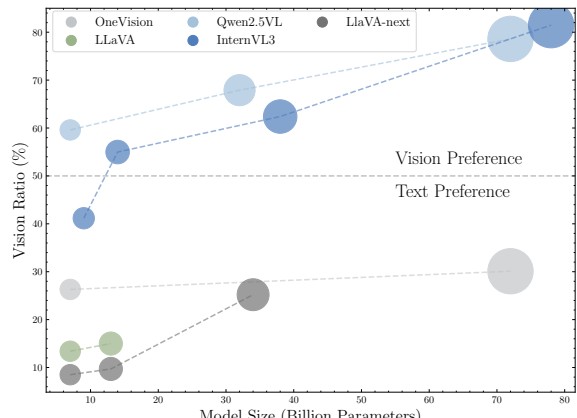

*Figure 2.* Results of modality preference across different MLLMs. **Left**: Quantified scores for **Vision**, **Others** and **Text** modalities using $S_{\text{vision}}$, $S_{\text{others}}$ and $S_{\text{text}}$ as well as Vision Ratio. Most MLLMs exhibit a strong text preference, while Qwen2.5VL and InternVL3 shows a certain degree of visual preference. **Right**: Trends of Vision Ratio with respect to model parameter size. For all model families, visual preference increases with model size.

choice format. To further reduce potential *position bias* in multi-modal inputs, we adopt a *consistent evaluation* strategy, similar to Liu et al. (2024d). Concretely, for each question, we construct two versions by swapping the order of the answer choices. A model's prediction is regarded as *consistent* only if it selects the same answer in both versions for a sample; otherwise, it is labeled as *inconsistent*. Such inconsistent samples are discarded from the subsequent measurement of modality preference.

**Evaluation Metric.** Inspired by prior work on evaluating stylistic or knowledge-related preferences of LLMs and MLLMs through conflict-pair contexts (Li et al., 2024b; Xie et al., 2023; Liu et al., 2025), we extend this to evaluate modality preference by designing a metric that captures how MLLMs respond to conflicting signals from different modalities. More importantly, through the careful design of the benchmark, we establish as a basis that the model can reliably understand both modalities in isolation. As shown in Table 18 and 19 in Apdx, all MLLMs achieve over 95% accuracy, provided with either textual or visual context.

Building on this, our metric evaluates modality preference by assessing how the model's responses align with textual or visual input when the two provide conflicting signals. The model's response is then categorized based on which modality it aligns with: 1) **Vision**: the response aligns with the visual context; 2) **Text**: the response aligns with the textual context; 3) **Others**: the response is uncertain, or inconsistent with either modality, which is discarded from further analysis. Then, we naturally define the **Vision Ratio** to quantify the model's preference toward the vision modality, defined as: $S_{\text{vision}}/(S_{\text{vision}} + S_{\text{text}})$, where $S_{\text{vision}}$ or $S_{\text{text}}$ denotes the score of the vision or text modality, computed as the proportion of samples whose responses are

categorized as **Vision** or **Text** across the dataset. $S_{\text{others}}$ is the proportion of samples whose responses are categorized as **Others**. Vision Ratio greater than 0.5 indicates that the model tends to favor visual context over text.

## 4. Modality Preferences in MLLMs

This section systematically investigate modality preference, structured around four key research questions: *1) Which modality do MLLMs prioritize? 2) What factors drive these preferences? 3) Can the Vision Ratio provide guidance for downstream task performance? 4) Can modality preference be controlled?* This investigation helps uncover the underlying mechanisms of modality preference and leverage these insights to downstream tasks.

### 4.1. Which Modality Do MLLMs Prioritize?

We use MC$^2$ to evaluate the modality preferences of 20 open-source MLLMs and the proprietary ChatGPT-4o-mini (Hurst et al., 2024a), detailed in Apdx C.1.

**Different MLLMs exhibit different modality preferences.** As described in Section 3.3, we quantify modality preference using **Vision Ratio** in the left panel of Figure 2 and detailed in Table 15. We observe that all MLLMs exhibit clear modality preference, with most models showing a strong preference for text; for instance, LLaVA1.5-7B attains only a 13.4% Vision Ratio. This aligns with the previous findings that MLLMs suffer from a severe language prior (Lee et al., 2024; Parcalabescu & Frank, 2024; Wu et al., 2025). Interestingly, Qwen2.5VL and InternVL3 show a certain degree of visual preference. To further rule out the potential influence of LLM-generated textual contexts, we conduct a manual re-annotation study. The results in Table 5

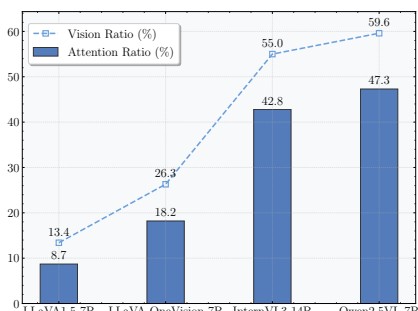 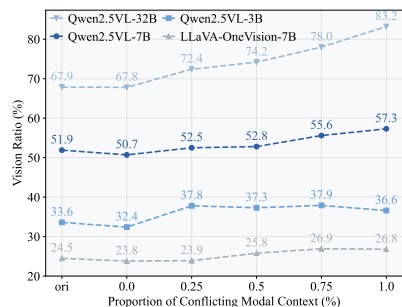 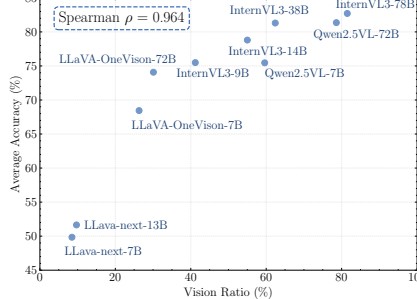

*Figure 3.* Analysis of modality preferences. **Left**: Vision Ratio closely aligns with Attention Ratio, suggesting that attention distribution for different modalities leads to distinct modality preference. **Middle**: Increasing the proportion of multi-modal conflict-context training data or training larger MLLMs yields larger preference shift. **Right**: Strong positive association between visual understanding ability (quantified as the average accuracy across seven widely used benchmarks) and the modality preference (measured by Vision Ratio) demonstrates that Vision Ratio provides a reliable indicator of visual understanding.

in Apdx. B.2 show that the Vision Ratio remains highly stable after replacing the original textual contexts with manually annotated ones. Besides, we conduct a sensitivity analysis on the number of MC$^2$ samples in Appendix E.1, showing that the current dataset scale is sufficient to obtain stable evaluation results for modality preference.

**Larger MLLMs exhibit stronger visual preferences.** We evaluate models from the LLaVA1.5, LLaVA-Next, Qwen2.5VL, InternVL3, and LLaVA-OneVision families to investigate the relationship between model size and modality preference. As shown in the right panel of Figure 2, we observe that for all model families, the preference for the vision modality increases with the model size. And Qwen2.5VL and InternVL3 models exhibit a significant preference for the vision modality once the model size increases. However, other models maintain a noticeable textual preference as model sizes increase.

**Vision Ratio serves as a reliable automated proxy for human assessments of an MLLM's modality preference.** We further verify whether the Vision Ratio can serve as a human-level measure for modality preference of MLLMs. We randomly sample 100 instances from MC$^2$ and compute the Vision Ratio of four MLLMs—Qwen2.5VL-7B, OneVision-7B, InternVL3-14B, and LLaVA1.5-7B. In addition, we craft prompts to elicit explicit reasoning chains from the models, specifically targeting their reliance on visual or textual information. To ensure labeling reliability, three expert annotators independently annotate the expressed modality preference for each response, with the final label determined by majority vote. The automatically obtained Vision Ratio scores (56.3%, 24.6%, 52.3%, 13.9%) are highly consistent with those given by human annotators (61.0%, 22.0%, 51.0%, 16.0%), with an average discrepancy of only 2.68%. This indicates that the Vision Ratio can act as a reliable, automated proxy for human assessment for modality preference.

### 4.2. What Factors Drive These Preferences?

Given that different MLLMs display varying modality preferences, we examine two primary sources: *the internal attention distribution* and *the training factors*.

**Different allocation of attention across modalities.** We compute the mean attention scores over all token positions from both modalities and define the ratio of visual attention to total attention as the **Attention Ratio**. By analyzing Qwen2.5VL-7B and LLaVA-OneVision-7B, we observe that the trends of the Attention Ratio closely align with the Vision Ratio across models in the left panel of Figure 3. This alignment suggests that MLLMs distribute attention unevenly between modalities, which in turn contributes to their divergent modality preferences.

**Impact of model scale and training data recipe.** MLLMs are typically initialized from LLMs with inherent text preference, and then adopt vision-centric training tasks to acquire vision understanding ability. Through reviewing the technical reports of the evaluated MLLMs, we find that they all adopt a common architecture comprising a vision encoder, an alignment layer, and an LLM. Thus, we hypothesize that the observed preferences mainly arise from two factors: 1) Exposure to more multi-modal contexts, especially with conflicting cases, drives more pronounced shifts in modality preference; 2) Larger LLMs are more capable of shifting their preference during training.

To examine these hypotheses, we construct a training dataset containing vision–text conflict contexts and fine-tune Qwen2.5VL-32B/7B/3B and LLaVA-OneVision-7B with varying proportions of samples with multi-modal conflict contexts to adjust preferences toward text or vision. We optimize MLLMs in the opposite direction of their original preferences and measure changes using the Vision Ratio. As shown in the middle panel of Figure 3, increasing the proportion of multi-modal contexts consistently leads to larger preference shifts, supporting

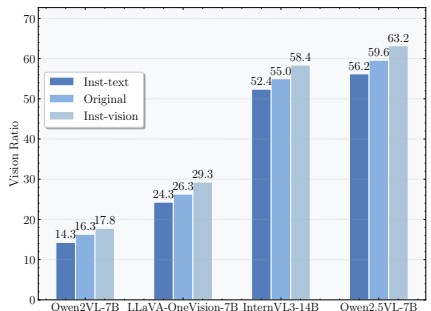 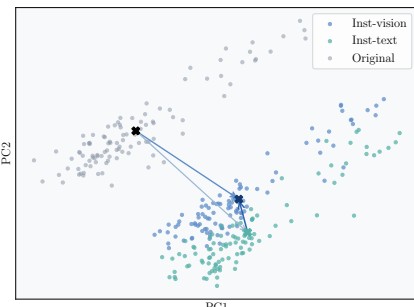 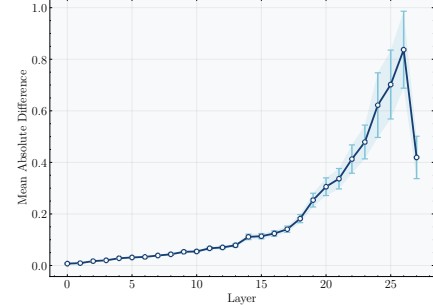

*Figure 4.* Analysis of modality preference under instruction-guidance. **Left**: Quantitative analysis shows that the expression of modality preference can be adjusted through instruction design, for example: "Follow the text context (**Inst-text**) or vision context (**Inst-vision**) to answer the question". **Middle**: Representation shifts quantified via PCA analysis under instructional interventions confirm that modality preference can be captured within the latent representation space. **Right**: Layer-wise absolute difference and standard deviation of hidden states between different instructions.

**Hypothesis 1**. This suggests that multi-modal inputs may create more challenging conditions, leading to greater preference shifts. Furthermore, Qwen2.5VL-32B exhibits greater shifts than Qwen2.5VL-7B/3B under the same training data, supporting **Hypothesis 2**. This indicates that larger LLMs demonstrate stronger adaptability for adjusting modality preference. Overall, Qwen2.5-VL's training frequently contains integrated vision-and-text contexts, that trigger cross-modal contention, forcing the model to resolve inconsistencies by prioritizing visual cues. In contrast, MLLM like LLaVA-OneVision-7B is trained on vision-as-context tasks, thus retaining more of their initial text preference. Larger MLLMs exhibit superior adaptability in shifting modality preferences. When optimized for vision understanding, larger models are more likely to achieve a more pronounced vision preference.

### 4.3. Can the Vision Ratio provide guidance for downstream task performance?

As a foundational behavioral prior, the identified modality preference can inform how a model integrates information across modalities. As such, the findings can offer relevant insights into the model's behavior in deeper cross-modal understanding tasks. To demonstrate the correlation between the modality preference and performance on downstream tasks, we evaluate the visual understanding abilities of 10 representative MLLMs. Specifically, we compute the average accuracy across 7 widely used benchmarks, as detailed in Apdx C.2. We then compare the visual understanding abilities with their modality preference measured by Vision Ratio using $MC^2$, as shown in the right panel of Figure 3. The results reveal a strong positive association between Vision Ratio and visual understanding ability across all MLLMs. Notably, Spearman's rank correlation (Sedgwick, 2014) reaches $\rho = 0.964$, demonstrating that Vision Ratio provides a highly reliable indicator of visual understanding ability.

### 4.4. Can Modality Preference Be Controlled?

We investigate whether modality preference can be controlled, and conduct latent space representation analysis to examine the underlying mechanisms. The detailed setting and results are provided in Apdx C.3.

**Expression of modality preference can be guided through instruction design.** We investigate the impact of instruction design on modality preference by explicitly directing the model to rely on a specific modality when answering a question. Specifically, we evaluate the modality preference using Vision Ratio for four representative MLLMs including Qwen2.5VL-7B, OneVision-7B, Qwen2VL-7B and InternVL3-14B under the text and vision preference instruction (Inst-text and Inst-vision). As shown in the left panel of Figure 4, instructions, probing the models toward any modality effectively shape their modality preferences.

**Modality preference direction in representation space.** To further understand how the intervention methods influence modality preference internally, we analyze the hidden representations of the models. Specifically, we apply Principal Component Analysis (PCA; Bro & Smilde, 2014) to the hidden states to identify the dominant direction of modality preference. The middle panel of Figure 4 shows that instruction-based interventions drive clear shifts in representations, revealing that the model's internal states are sensitive to modality control. This motivates us to develop specific methods to adjust modality preference.

## 5. Method

Inspired by Section 4.4, we propose to use representation engineering (Zou et al., 2023; Li et al., 2026; 2025) to steer modality preference. As shown in Figure 5, the proposed framework consists of Modality Preference Probing (§5.1) and Modality Preference Steering (§5.2).

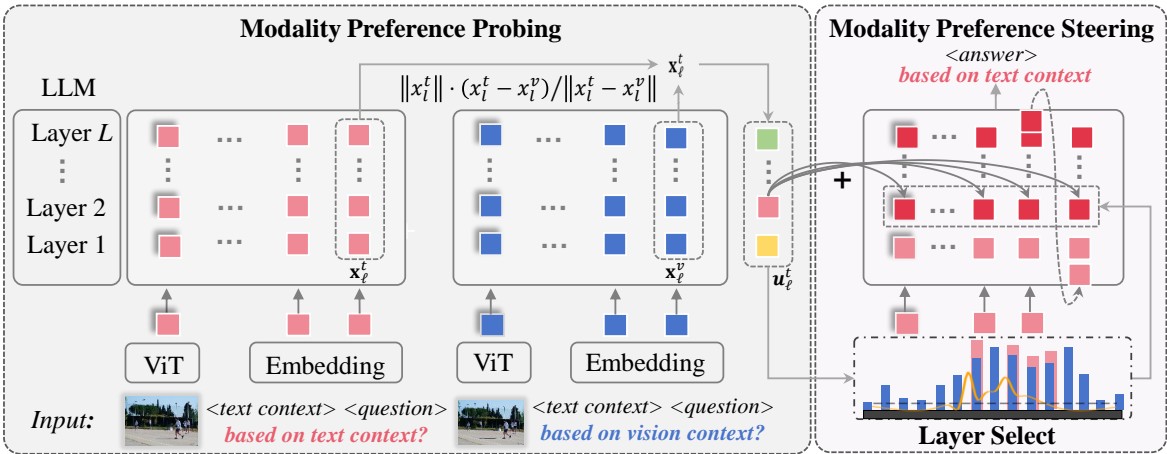

*Figure 5.* Overall framework of the proposed method. Modality Preference Probing collects the neural activity, computes and scales the direction of modality preference. Modality Preference Steering selects the target layer during the second inference and adds the scaled modality preference direction to the representation at the corresponding layer at each inference step.

*Table 1.* Performance for steering Qwen2VL-7B and OneVision-7B towards vision modality, and Qwen2.5VL-7B and InternVL3-8B towards text modality, measured by $S_{\text{vision}}$ and $S_{\text{text}}$.

| Preference | Model | MLLM | InstDesign | CoT | FewShot | Ours |
|---|---|---|---|---|---|---|
| **Text↑** | Qwen2.5VL-7B | 35.4 | 37.7 | 55.6 | 61.1 | **63.6** |
| | InternVL3-8B | 20.9 | 31.6 | 36.7 | 38.2 | **42.8** |
| **Vision↑** | Qwen2VL-7B | 15.3 | 32.3 | 34.2 | 17.2 | **48.1** |
| | OneVision-7B | 37.5 | 52.8 | 53.1 | 49.8 | **57.1** |

*Table 2.* Performance of the proposed method on the reasoning tasks (MathVista, TallyQA and VSR) measured by accuracy for Qwen2.5VL-7B and multi-modal translation task (Ambigcaps) measured by BLEU scores for English (En) ↔ Turkish (Tr).

| Dataset | CoT | InstDesign | Ours |
|---|---|---|---|
| MathVista | 50.0 | 59.0 | **62.3** |
| TallyQA | 61.6 | 74.4 | **76.6** |
| VSR | 45.6 | 51.3 | **53.2** |
| Ambigcaps (**En→Tr**) | 9.03 | 9.45 | **10.2** |
| Ambigcaps (**Tr→En**) | 18.63 | 18.98 | **19.89** |

## 5.1. Modality Preference Probing

We probe and collect neural activity that represents the direction of modality preference. Inspired by the pre-training Next Token Prediction objective of decoder-only MLLMs (Hurst et al., 2024a) and the method to extract classification features (Feucht et al., 2024), we collect neural activity from the last token in the input text. The process involves probing modality preference through two requests: one with a vision preference probing (e.g., 'answer the question based on the vision context') and another with a text preference probing (e.g., 'answer the question based on the text context'). Let us denote these inputs by $q^v$ and $q^t$, and consider a set of $N$ such pairs $(q_i^v, q_i^t)$, $i \in \{1, \ldots, N\}$. Let $\mathbf{x}_{i,\ell}^v, \mathbf{x}_{i,\ell}^t \in \mathbb{R}^d$ be the hidden states for the two queries at the last token at layer $\ell \in \{1, \ldots, L\}$, where $d$ is the dimension of the chosen MLLM. We identify the direction of modality preference by computing the difference in the hidden states between the paired inputs. Formally, we compute a vector $\mathbf{u}_\ell \in \mathbb{R}^d$ representing the direction towards the text modality at layer $\ell$ for a given query as:

$$\mathbf{u}_\ell^t = \frac{1}{N} \sum_{i}^{N} \left( \mathbf{x}_{i,\ell}^t - \mathbf{x}_{i,\ell}^v \right). \quad (1)$$

Averaging over different queries allows us to capture the

activation values most closely associated with modality preference, independent of questions. As shown in the right panel of Figure 4, we compute the absolute values and standard deviations of the modality preference direction $\mathbf{u}_\ell^t$ across different samples. We observe that layers 20–23 exhibit higher absolute values and lower variance, indicating that the preference direction is more prominent and stable in these layers. Based on this observation, we select the corresponding layer $\ell'$ of the model to control the direction of modality preference in Section 5.2. We provide the reults of similar patterns for Qwen2VL-7B, Qwen2.5VL-7B, LLaVA-OneVision and InternVL3 and conduct a mechanistic logit-lens analysis to further justify layer selection in Apdx E.5.4.

## 5.2. Modality Preference Steering

After obtaining the probing direction vector, we compute the steering vector by re-scaling the vector $\mathbf{u}_\ell^t$ with a weight $w$. The scaling process must carefully balance two objectives: 1) it must be strong enough to effectively steer the model's modality preference, 2) it must preserve the model's normal output behavior. In our preliminary experiments, we observe that setting the weight too large

*Table 3.* Performance of the proposed method on the visual understanding benchmark, Phd (Liu et al., 2024c). we report the accuracy results on the phd-icc/phd-iac.

| Model | Attribute | Sentiment | Positional | Counting | Object | Avg |
|---|---|---|---|---|---|---|
| Qwen2VL-7B | 10.0 / 28.5 | 2.5 / 8.5 | 3.5 / 20.5 | 6.0 / 30.5 | 8.0 / 50.0 | 6.0 / 27.6 |
| +InstDesign | 14.5 / 34.5 | 2.5 / 13.0 | 1.5 / 26.0 | 5.5 / 39.0 | 25.0 / 60.0 | 9.8 / 34.5 |
| +CoT | 5.0 / 15.5 | 6.0 / 23.5 | 8.5 / 30.2 | 6.5 / 17.0 | 40.5 / 59.0 | 13.3 / 29.0 |
| +FewShot | 3.0 / 17.0 | 0.5 / 9.0 | 1.5 / 14.5 | 5.0 / 29.0 | 2.0 / 37.0 | 2.4 / 21.3 |
| +Ours | 10.0 / 34.4 | 11.0 /16.5 | 14.0 / 28.3 | 5.0 / 37.4 | 51.5 / 64.0 | **18.4** / **36.1** |
| OneVision-7B | 11.5/ 20.5 | 1.5 / 5.0 | 1.5 / 16.5 | 6.5 / 28.5 | 11.0 / 52.0 | 6.4 / 24.5 |
| +InstDesign | 16.0 / 27.0 | 5.5 / 12.5 | 6.0 / 31.5 | 13.5 / 30.5 | 34.0 / 61.5 | 15.0 / 32.6 |
| +CoT | 17.3 / 28.4 | 6.2 / 12.9 | 7.8 / 33.2 | 13.8 / 30.9 | 34.5 / 62.1 | 15.9 / 33.1 |
| +FewShot | 17.0 / 28.0 | 6.0 / 13.0 | 7.2 / 32.8 | 13.9 / 31.0 | 34.8 / 62.3 | 16.2 / 33.4 |
| +Ours | 19.6 / 30.5 | 7.8 / 13.5 | 10.3 / 36.4 | 15.1 / 29.8 | 35.6 / 63.5 | **17.7** / **34.7** |

leads to repetitive and meaningless outputs, whereas a too small weight fails to obtain any noticeable change for modality preference. Unlike previous approaches (Zou et al., 2023; Stolfo et al., 2024) that rely on exhaustive search over a validation set to determine the weight, we propose a principled method that aligns the mean of the probed direction distribution with the mean of the original distribution of hidden states. This strategy ensures that the steering remains effective without disrupting the model's inherent generation capabilities. Formally, the weight is determined by aligning the mean of the probed direction with the central distribution of the original hidden states and the steering vector is computed by:

$$\mathbf{s}_\ell^t = w\mathbf{u}_\ell^t, \text{ where } w = \frac{1}{N}\sum_{i=1}^{N}\frac{\|\mathbf{x}_{i,\ell}^t\|}{\|\mathbf{u}_\ell^t\|} \qquad (2)$$

Finally, during the second inference, we adjust the hidden states at the selected layer $\ell'$ at each decoding step by adding $\mathbf{s}_{\ell'}^t$ to all tokens to steer the model's response toward text modality. Similarly, steering towards vision modality is performed towards the opposite direction and no additional data or labels are introduced. Unlike token-wise steering in ICV (Liu et al., 2023b), we apply a unified steering vector (identical strength and direction) to both vision and text tokens. As noted by Zheng et al., 2025, vision and text representations in MLLMs are often out-of-distribution relative to each other. Consequently, applying disparate steering vectors maybe lead to a significant distortion of the relative distribution between vision and text modalities, thereby undermining the model's intrinsic capabilities.

## 5.3. Experiments

We evaluate the proposed method for controlling modality preference on **MC²** and multiple downstream tasks across four representative MLLMs: Qwen2VL-7B, Qwen2.5VL-7B, OneVision-7B, and InternVL3-8B. We compare our approach against several widely-used training-free baselines: *MLLM* (zero-shot direct response), *InstDesign* (instruction-based guidance for modality preference), *CoT* (utilizing intermediate processing steps), and *FewShot*

(four-shot in-context learning). Implementation details are provided in Apdx D. As shown in Table 1, our method consistently outperforms all baselines on **MC²** across both evaluation settings. These results demonstrate the superior effectiveness of our approach in precisely adjusting modality preference compared to standard prompting techniques.

We further assess our method across a variety of downstream scenarios: 1) Multi-modal reasoning tasks on MathVista (Lu et al., 2023), TallyQA (Acharya et al., 2019) and VSR (Liu et al., 2023a); 2) Multi-modal understanding tasks on PhD (Liu et al., 2024c) and AmbigCaps (Li et al., 2021), a multi-modal machine translation (MMT) task. The Phd dataset includes two subsets—PhD-ica, which contains irrelevant textual context, and PhD-icc, which introduces misleading or incorrect textual information—both of which increase the risk of hallucination. As shown in Table 2, our method consistently surpasses both the *CoT* and *InstDesign* baselines. We observe that by steering the model toward a visual preference, our method increases reliance on the original visual input. This effectively prevents the final decision from being misled by potential vision hallucination that often emerges during the sequential steps of CoT. On the AmbigCaps translation task, our method achieves a significant improvement of 1.33 BLEU score over the strongest baseline. Furthermore, Table 3 demonstrates substantial improvements in multi-modal understanding. Specifically, when applied to Qwen2VL-7B, our method surpasses the best-performing baseline by an average margin of 6.1 percentage points. In addition, experiments on Qwen2-VL-7B show that applying token-wise steering with ICV leads to significant performance degradation, with scores of 13.6 on Phd-icc and 30.1 on Phd-iac. Additional results on more tasks and models are provided in Apdx E.3.

## 5.4. In-depth Analysis for Steering Method

To investigate the internal mechanism of the steering method, we analyze the attention scores of the generated tokens toward the vision and text contexts using the proposed steering method and the InstDesign method with the samples from $MC^2$. In Figure 6, we visualize the attention distribution at the 24th layer by steering the Qwen2.5VL-7B at the 22nd layer towards the text modality using the case in Figure 13. We observe that after applying the steering method, the model's attention weight toward the text modality significantly increases. This change clearly demonstrates that the steering mechanism successfully alters the modality preference by enhancing the model's dependency on the text modality. We further analyze generation quality using the proposed method in Apdx. E.4 and provide more ablation experiments, the analysis of the latency, memory usage and the prerequisites for the proposed method in Apdx E.5.

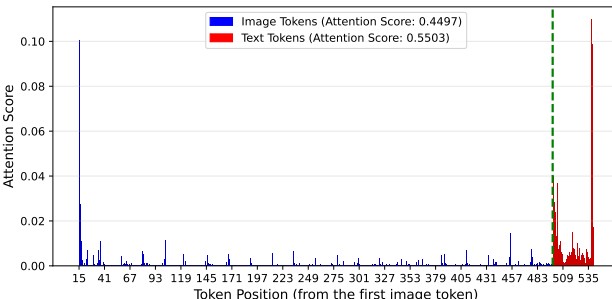 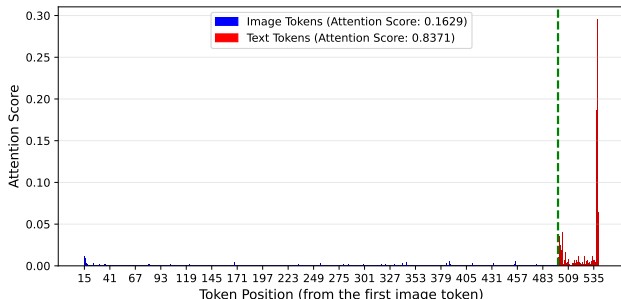

*Figure 6.* Visualization of attention scores toward multi-modal contexts. We compare InstDesign (Baseline; Left) with the proposed method (Right) at the 24-th layer (steered at the 22-nd layer). The results demonstrate that our method significantly enhances attention weights toward the targeted text modality, effectively recalibrating the model's internal modality preference by increasing textual dependency.

# 6. Discussion

Our study shows that modality preference is a systematic property of MLLMs, with implications beyond benchmark performance. It provides a lens for understanding how models arbitrate multimodal evidence, and also offers practical signals for diagnosing and improving multimodal training for MLLMs. We discuss these potential implications from two perspectives.

**"Optimal" modality preference should be understood from both intrinsic and task-dependent perspectives.** From an intrinsic alignment perspective, a Vision Ratio close to $0.5$ is desirable, as it reflects modality-neutral behavior where the model does not systematically prioritize visual or textual evidence. Such neutrality is important for faithful multimodal assistants, particularly in dialogue and retrieval-augmented generation scenarios where over-reliance on a modality prior may cause evidence-inconsistent responses. Nevertheless, modality-neutrality does not necessarily correspond to the best downstream performance for existing benchmarks. The performance-optimal Vision Ratio depends on the evidence reliance of each specific task: vision-centric benchmarks benefit from stronger visual reliance, whereas text-dominant tasks such as AmbigCaps favor lower Vision Ratios. Thus, Vision Ratio as $0.5$ should be viewed as a desirable indicator of intrinsic modality-neutrality, while the best-performing modality preference remains task-dependent.

**Modality preference offers practical value for model diagnosis and training.** Vision Ratio can serve as a lightweight proxy for assessing visual understanding for large-scale evaluation. In omni-modal training, it may further indicate whether a model preserves existing modality capabilities while learning new ones. More broadly, systematic modality preference reveals potential imbalance in multimodal optimization, suggesting that future omni-modal models may benefit from more balanced training objectives that explicitly regulate cross-modal reliance.

# 7. Conclusion

This paper investigates modality preference in multi-modal large language models (MLLMs). We carefully curate a modality conflict dataset and use a controlled experimental setup to quantitatively evaluate modality preference. Besides, we find the direction of modality preference can be captured within the latent representations of MLLMs. Inspired by this, we propose a modality preference probing and steering method, which enables significant and flexible changes in modality preference. Experiments show that the proposed method generalizes well to downstream tasks, such as multi-modal reasoning and understanding tasks.

# Impact Statement

The proposed MC$^2$ evaluates the modality preference of MLLMs. Understanding which modality a model prioritizes could be used to circumvent safety mechanisms (e.g., hiding harmful content in the favored modality), making it harder for filters to detect inappropriate content. Therefore, it is essential to incorporate effective safeguards in MLLMs to filter out any inappropriate materials.

# Acknowledgments

This work was supported in part by the National Science and Technology Major Program (2024ZD01NL00101), in part by the Science Fund for Creative Research Groups of the National Natural Science Foundation of China under Grant 62521006, in part by the National Natural Science Foundation of China (62406091, 62276077, U23B2055, 62350710797, 62506182), in part by the Xinjiang Science and Technology Development Plan for the Two Innovation Demonstration Zones along the Silk Road Economic Belt under Grant 2024LQ03003, in part by Guangdong S&T Program (2024B0101050003), in part by the Guangdong Basic and Applied Basic Research Foundation (2026A1515011718, 2024A1515011205), in

part by Shenzhen Science and Technology Program (KQTD20240729102154066) and in part by the Major Key Project of PCL (PCL2025A03).

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

All codes, data, and instructions for our MC$^2$ can be found in `https://github.com/EchoDreamer/Modality-Preference`. MC$^2$ is released under a Creative Commons Attribution 4.0 License (CC BY 4.0).

Our supplementary materials are summarized as follows:

- Appendix A: Discussion including Limitations, Use of LLM and License of Assets.

- Appendix B: Dataset Construction

- Appendix C: Model Evaluation

- Appendix D: Method Applying

- Appendix E: More Experiment Analysis

## A. Discussion

### A.1. Limitations

This paper investigates the modality preference in multi-modal large language models (MLLMs) using a controlled experiment setup with a modality conflict dataset. In constructing the dataset, we employs LLaVA1.5-7B and QwenVL-7B to filter samples and ensure that most models could answer questions correctly based on a single modality. However, this process requires multiple iterations and turned out to be time-consuming. Therefore, devising a more efficient and elegant method for sample selection may be of greater importance.

### A.2. Use of LLM

In this work, we use the LLMs including GPT-4o-mini (Hurst et al., 2024a) and DeepSeekV3 (Liu et al., 2024a), and MLLMs including LLaVA1.5-7B (Liu et al., 2024b) and QwenVL-7B (Bai et al., 2023) to help annotate the text context in MC$^2$, as detailed in Section B. We evaluate the modality preference of 20 open-source MLLMs and GPT-4o-mini (Hurst et al., 2024a), and steer the modality preference of Qwen2VL-7B (Wang et al., 2024), Qwen2.5VL-7B (Bai et al., 2025), and OneVision-7B (Li et al., 2024a) and InternVL3-8B (Zhe et al., 2024). Besides, we also utilize the LLMs to correct the grammatical errors.

### A.3. License of Assets

All images in MC$^2$ are publicly available from COCO (Lin et al., 2014). We release our benchmark under a Creative Commons Attribution 4.0 License (CC BY 4.0) to enhance global accessibility and foster innovation and collaboration in research.

## B. Dataset Construction

### B.1. Conflict Text Context Generation

**Details for data generation using LLMs** To ensure reproducibility and transparency, we include the exact prompts used in our data generation process. These prompts were designed to generate the candidate textual contexts and corresponding answers using GPT-4o-mini (Hurst et al., 2024a) and DeepSeekV3 (Liu et al., 2024a). Below, we provide representative examples of the prompts used during dataset construction given the caption of an image, question, the answer for the question based on image and the task type for the question. For the full list of prompts, please refer to the project repository.

**Human Verification** Although the text contexts and answers generated by strong LLMs—filtered through judge MLLMs such as LLaVA1.5-7B (Liu et al., 2024b) and QwenVL-7B (Bai et al., 2023)—generally yield reliable results, we further incorporate manual inspection to ensure the high quality of data annotations. Specifically, we verify that the visual and textual contexts are indeed in conflict, and that each modality independently supports the corresponding answer to the given question. This involves a two-stage manual review process:

- **Modality-Answer Alignment.** First, for each context from different modalities (image and text), annotators assess whether it independently provides sufficient information to correctly answer the question. This step is particularly

important because the original VQA answers in the TDIUC (Kafle & Kanan, 2017) dataset may contain error annotations, and the LLM-generated contexts and answers may occasionally be inconsistent.

- **Conflict Verification.** Next, annotators examine whether the visual and textual contexts are semantically inconsistent with respect to the question. That is, the two modalities should lead to different correct answers when considered separately. Samples where both modalities lead to the same answer are discarded, as they do not reflect a true modality conflict.

Samples that do not meet either verification criterion are flagged for further review. Depending on the nature and severity of the issue, we take one of the following actions: revise the prompt to improve clarity, regenerate the problematic part of the sample (e.g., the question or context), or discard the sample entirely if it cannot be reasonably corrected.

To ensure consistency and reduce subjectivity, each category (i.e., vision-aligned, text-aligned, and conflict) is independently verified by three trained annotators. Disagreements are resolved through discussion or majority voting. In addition, we conduct random spot-checks throughout the dataset to ensure the consistency and reliability of the annotations.

---

**Conflict Context Generation for counting task using DeepSeekV3**

Instruction:
# Given a description of an image and a corresponding counting type question with its answer, now you are required to generate a text context that points to an answer that fluctuates by 1 or 2 from the original answer. The context explicitly supports the new answer, providing clear evidence that aligns logically with the counting question. Only one alternative answer should be generated.
Caption: {**caption**}
Question: {**question**}
Answer: {**answer based on vision context**}
Output the new answer enclosed in <answer> </answer> and the context enclosed in <context> </context> tags.

---

**Conflict Context Generation of for other tasks using DeepSeekV3**

Instruction:
# Given the caption of an image and a corresponding {**task-type**} type question with its answer, now you are required to generate a text context as a premise that supports a new distractor answer for the question. The context should mimic the environment described in the caption but should not include {**answer based on vision context**}, while maintaining logical consistency within the context. Only one alternative answer should be generated.
Caption: {**caption**}
Question: {**question**}
Output the answer enclosed in <answer> </answer> and the context enclosed in <context> </context> tags.

---

**Conflict Context Generation for other tasks using GPT-4o-mini**

Instruction:
# Given a caption of an image and a corresponding counting question with its answer, you are required to generate a single text context that provides an indirect premise leading to a new answer that fluctuates by 1 or 2 from the original answer. The context should build an indirect premise to the new answer. Carefully design this context. For this task, I want you to first describe the scene with a certain quantity and then introduce an increase or decrease in that quantity to imply the final answer and don't include the final answer. Only one alternative answer should be generated.
Caption: {**caption**}
Question: {**question**}
Answer: {**answer based on vision context**}
Task-type: {**task-type**}
Output the answer enclosed in <answer> </answer> and the context enclosed in <context> </context> tags.

---

Conflict Context Generation for count task using GPT-4o-mini

Instruction:
# Given the caption of an image and a corresponding question with its answer, now you are required to generate a text context as the indirect premise of a new answer for the question, which belongs to the same category as the original answer. The context should support the new answer, include the caption while maintaining logical consistency within the context and don't include the final answer. Only one alternative answer should be generated.
Caption: {**caption**}
Question: {**question**}
Answer: {**answer**}
Task-type: {**task-type**}
Output the new answer enclosed in <answer> </answer> and the context enclosed in <context> </context> tags.

---

*Table 4.* Average text context length across different task types in the MC$^2$ dataset.

| statistics | Sport | Attribute | Sentiment | Positional | Counting | Color | Activity | Object | Avg |
|---|---|---|---|---|---|---|---|---|---|
| **Text Length** | 52.48 | 33.50 | 39.69 | 31.53 | 37.12 | 31.15 | 49.68 | 39.71 | 39.36 |

### B.2. Manual Re-annotation for Benchmark–Metric Coupling

To directly examine whether the Vision Ratio is affected by potential artifacts introduced by LLM-generated textual conflicts, we conduct an additional manual re-annotation study. Specifically, we manually re-annotate the textual contexts in $MC^2$ while keeping the corresponding images unchanged, and then re-evaluate a broad set of MLLMs on the revised benchmark.

**Rigorous manual annotation.** For each original sample, the conflicting text context is independently rewritten by three annotators. Annotators are instructed to preserve the original scene semantics, writing style, and non-target details as much as possible, while only modifying the answer-relevant evidence such that the text supports the conflicting answer. The final text context is determined by majority agreement and further manually verified to ensure fluency, grammatical correctness, and semantic consistency with the image, except for the intended cross-modal conflict.

**Large-scale re-evaluation.** Following the same evaluation protocol as the original $MC^2$, we first verify that the evaluated models still achieve near-perfect unimodal text understanding accuracy, around $99\%$, on the manually annotated version. We then re-evaluate multiple MLLMs using the manually annotated benchmark. As shown in Table 5, the Vision Ratio remains highly stable across models after manual re-annotation. Most models exhibit only minor changes, and the overall trends are consistent with those observed on the original benchmark.

**Consistent results.** The results remain stable across models on the manually annotated benchmark, with most changes within $5\%$. Importantly, the relative trends among different MLLMs remain largely unchanged. These results suggest that the Vision Ratio is not primarily driven by artifacts specific to LLM-generated textual contexts, further supporting the reliability of $MC^2$ for evaluating modality preference.

### B.3. Data Statistics

We computed the average number of words in the text context for all samples within each task type using the `spaCy` library.[2] As shown in Table 4, while there are some variations in text length across tasks, the differences are relatively minor. This indicates that text length is unlikely to be a confounding factor in evaluating modality preference across different task types.

### B.4. Illustrative Samples from the MC$^2$ Benchmark

To provide an intuitive understanding of the MC$^2$ benchmark and the nature of modality conflict, we present a few representative samples covering different task types as shown in Figure 7, Figure 8, Figure 9 and Figure 10.

---

[2]We use the `spaCy` library in Python, available at `https://pypi.org/project/spacy/`.

**\<image\> is a placeholder for below image**

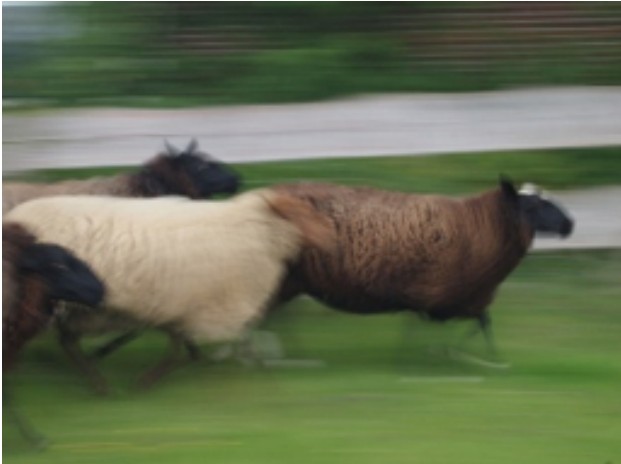

**\<image\> is a placeholder for below image**

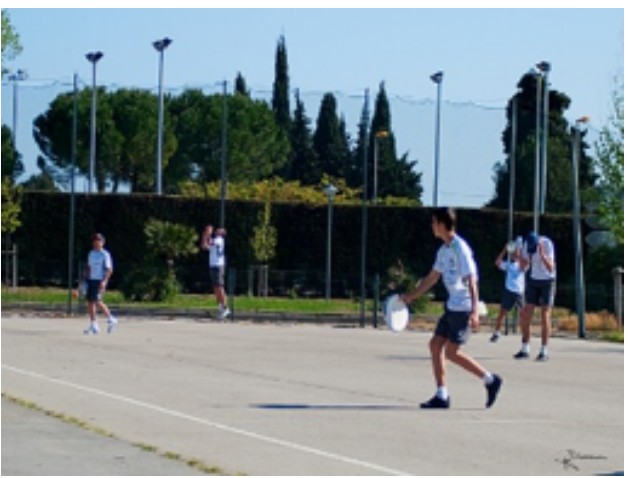

**User:** \<image\>**Conflict Text Context:** *Three sheep are peacefully eating grass*, surrounded by lush greenery. Their heads are lowered as they nibble on the fresh blades, completely undisturbed. **Question:** What are the cows in the back doing?
**Assistant:** \<output\>
**vision-based Answer:** *running*
**Text-based Answer:** *eating*

**User:** \<image\>**Conflict Text Context:** In the photo, there are three boys playing Frisbee, and one more boy is partially visible in the corner, bending down to tie his shoelaces, *making a total of four people*. **Question:** How many people are in the photo?
**Assistant:** \<output\>
**vision-based Answer:** *five*
**Text-based Answer:** *four*

*Figure 7.* Illustration of using modality context conflict pairs to investigate modality preference in activity recognition (Left) and counting tasks (Right). The highlighted areas indicate the points of conflict between visual and textual contexts.

**\<image\> is a placeholder for below image**

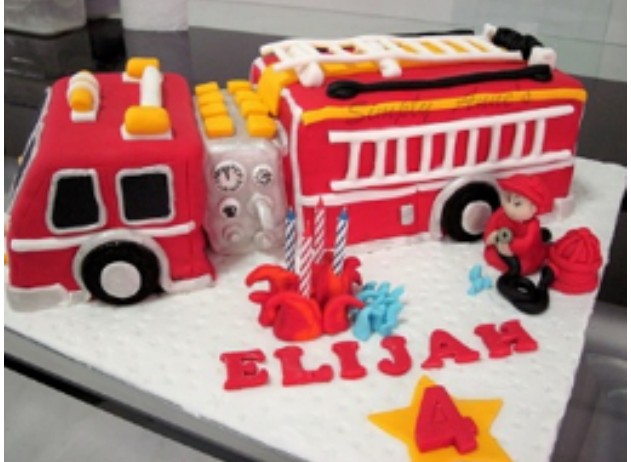

**\<image\> is a placeholder for below image**

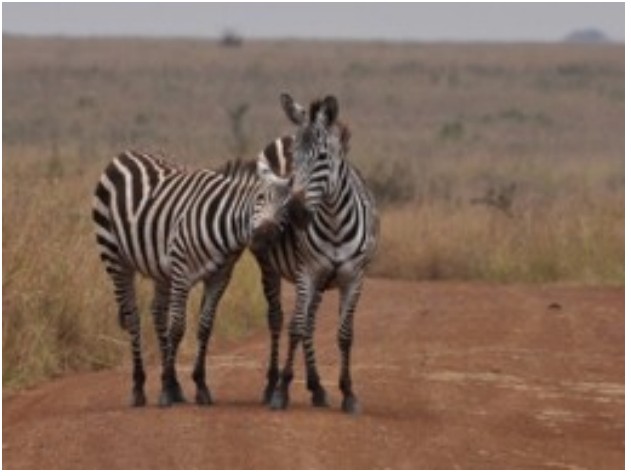

**User:** \<image\>**Conflict Text Context:** *The birthday cake was designed to look like a sleek police car*, complete with edible flashing lights and a fondant badge on the side. **Question:** What is the cake in the shape of?
**Assistant:** \<output\>
**vision-based Answer:** *fire truck*
**Text-based Answer:** *police car*

**User:** \<image\>**Conflict Text Context:** *Two wildebeests are standing in a dry*, grass-less savanna, their dark coats contrasting with the dusty ground. The area is sparse, with only a few scattered shrubs visible in the background. **Question:** What animal is shown?
**Assistant:** \<output\>
**vision-based Answer:** *zebras*
**Text-based Answer:** *wildebeests*

*Figure 8.* Illustration of using modality context conflict pairs to investigate modality preference in attribute recognition (Left) and object recognition tasks (Right). The highlighted areas indicate the points of conflict between visual and textual contexts.

**<image> is a placeholder for below image**

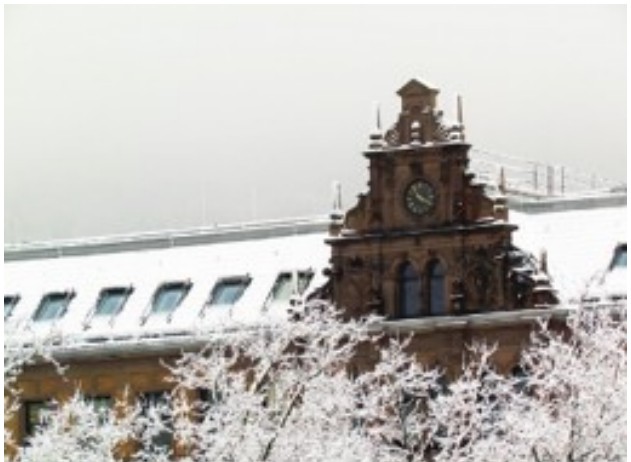

**User:** <image>**Conflict Text Context:** A large brown clock tower mounted in the face of *a building overlooks a vibrant park filled with lush green trees*. The contrast between the brown tower and the surrounding greenery creates a picturesque scene.
**Question:** What color are the trees?
**Assistant:** <output>
**vision-based Answer:** *white*
**Text-based Answer:** *green*

**<image> is a placeholder for below image**

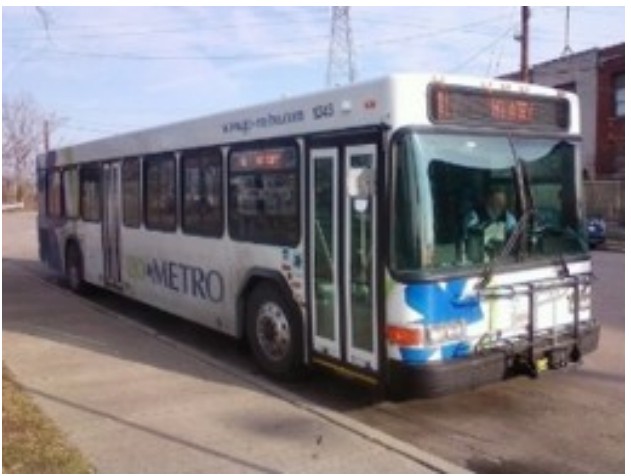

**User:** <image>**Conflict Text Context:** A white bus with a large rack on the front is parked by the beach, designed to *carry equipment for surfing* .The rack is sturdy and spacious, perfect for securing bulky items. **Question:** What can you hang on the rack on the front of the bus?
**Assistant:** <output>
**vision-based Answer:** *bikes*
**Text-based Answer:** *surfboards*

*Figure 9.* Illustration of using modality context conflict pairs to investigate modality preference in color recognition (Left) and positional reasoning (Right) tasks. The highlighted areas indicate the points of conflict between visual and textual contexts.

**<image> is a placeholder for below image**

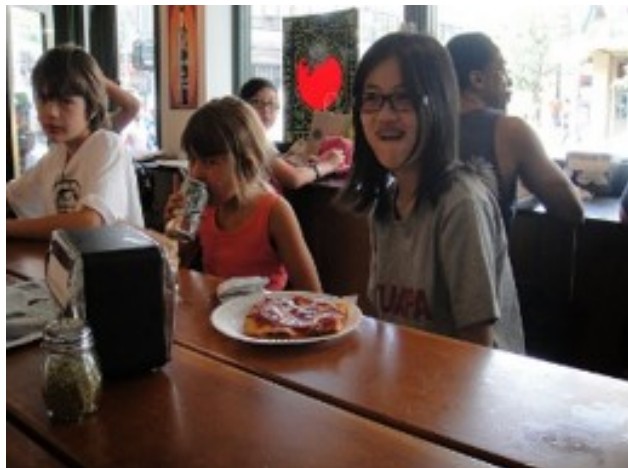

**User:** <image>**Conflict Text Context:** A girl sitting at a counter with a piece of pizza, staring blankly at the wall while the pizza grows cold in front of her. The room is quiet, and *she seems uninterested in her surroundings*. **Question:** What is the girl on the right feeling in the image?
**Assistant:** <output>
**vision-based Answer:** *happy*
**Text-based Answer:** *bored*

**<image> is a placeholder for below image**

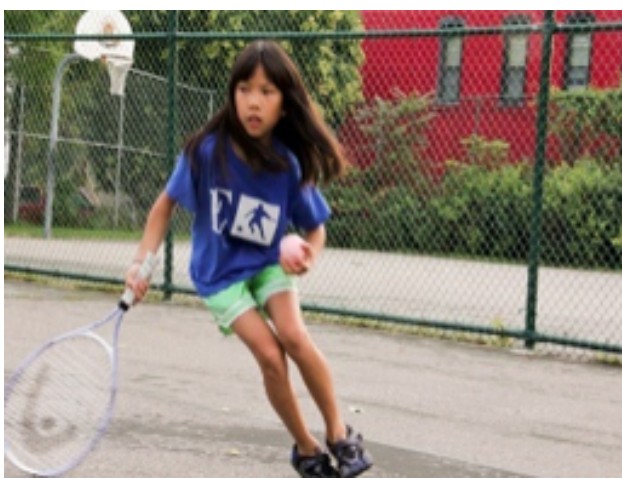

**User:** <image>**Conflict Text Context:** The young girl is running swiftly across the field, *dribbling a soccer ball with precision* as she maneuvers past imaginary opponents. Her focus is on scoring a goal, and she practices her footwork with determination.
**Question:** What sport is depicted in the picture?
**Assistant:** <output>
**vision-based Answer:** *tennis*
**Text-based Answer:** *soccer*

*Figure 10.* Illustration of using modality context conflict pairs to investigate modality preference in sentiment understanding and object recognition tasks. The highlighted areas indicate the points of conflict between visual and textual contexts.

*Table 5.* Comparison of Vision Ratio between the original $MC^2$ benchmark and the manually annotated version. The textual contexts are manually re-annotated while the images remain unchanged. Relative Change is computed with respect to the original Vision Ratio.

| MLLM | Original (%) | Manual-annotated (%) | Relative Change (%) |
| --- | --- | --- | --- |
| LLamaVision | 23.0 | 21.5 | -6.5 |
| LLaVA1.5-7B | 13.4 | 13.2 | -1.5 |
| OneVision-7B | 26.3 | 24.5 | -6.8 |
| Owl3-7B | 30.6 | 30.1 | -1.6 |
| Qwen2VL-7B | 16.3 | 16.0 | -1.8 |
| CogVLM2-19B | 34.3 | 35.4 | 3.2 |
| GLM-4V-9B | 36.6 | 37.5 | 2.5 |
| SPHINX-V2-1K | 25.1 | 25.3 | 0.8 |
| LLaVA-Next-7B | 8.5 | 8.5 | 0.0 |
| InternVL3-14B | 55.0 | 56.1 | 2.0 |
| Qwen2.5VL-7B | 59.6 | 62.1 | 4.2 |
| Qwen2.5VL-32B | 67.9 | 69.5 | 2.4 |
| InternVL3-38B | 62.4 | 62.0 | -0.6 |

## C. Model Evaluatiion

### C.1. Evaluatiion Detail for Modality Preference

We assess open-source multi-modal large language models (MLLMs) with different parameter sizes, including LLaVA1.5-7B/13B (Liu et al., 2024b), LLama3.2-11B-Vision-Instruct (Grattafiori et al., 2024), OneVision-7B/72B (Li et al., 2024a), CogVLM2-19B (Hong et al., 2024a), mPLUG-Owl3-24-07 (Ye et al., 2024), Qwen2VL-7B (Wang et al., 2024), GLM-4V-9B (Du et al., 2022), SPHINX-V2-1K (Lin et al., 2023), InternVL3-9B/14B/38B/78B (Zhe et al., 2024), LLaVA-next-7B/13B/34B (Liu et al., 2024b) and Qwen2.5VL-7B/32B/72B (Bai et al., 2025). All the open-source models are evaluated using NVIDIA A100 or A800 GPUs. We also evaluate the proprietary model, GPT-4o-mini (Hurst et al., 2024a) via the official API.

**Details of single-modality context evaluation** Before evaluating modality preference, we first assess the ability of MLLMs to answer questions accurately given a single-modality context in the $MC^2$ dataset. Specifically, we evaluate the models' accuracy in answering based on text context and based on vision context (based on the image). As shown in Table 18 and Table 19, all models achieve over 95% accuracy when provided with either textual or visual context. This indicates that question understanding and the understanding of single-modality context do not affect the modality preference evaluation. Therefore, we have excluded this confounding factor from the analysis.

**Details of results for modality preference evaluation** We provide the results of modality preference for several models in the left panel of Figure 2 in the main text. More detailed modality preference evaluation results are presented in Table 15.

### C.2. Can the Vision Ratio provide guidance for downstream task performance?

We evluate the performance of Qwen2.5VL-7B, Qwen2.5VL-72B, InternVL3-9B, InternVL3-14B, InternVL3-38B, InternVL3-78B, LLaVA-OneVison-7B, LLaVA-OneVison-72B, LLava-next-7B, LLava-next-13B on 7 general multi-modal understanding benchmarks including MMMU (Yue et al., 2024), MME (Chaoyou et al., 2023), MMBench (Liu et al., 2024d), RealwordQA (X.AI, 2024), MMStar (Masry et al., 2022), InfoVQA (X.AI, 2024) and ChartQA (Masry et al., 2022). We compute the average score on all datasets, where MME score is normalized between 0-1, as shown in Table 6.

### C.3. The Details for Controlling Modality Preference

**More results for controlling modality preference through instruction design.** In the left panel of Figure 4, we provide the Vision Ratio results for OneVision-7B, Qwen2.5VL-7B, Qwen2VL-7B and InternVL3-8B. We also present more results on controlling modality preference through instruction design for preference towards the vision modality and the text modality in Table 16 and Table 17. For each setting, we report the results measured by $S_{\text{vision}}$, vision-based accuracy and

| Model | MMMU | MME | MMBench | RealworldQA | MMStar | HallBench | InfoVQA | ChartQA | Avg | Vision Ratio |
|---|---|---|---|---|---|---|---|---|---|---|
| Qwen2.5VL-7B | 58.6 | 83.8 | 83.5 | 68.5 | 63.9 | 52.9 | 82.6 | 87.3 | 75.5 | 59.6 |
| Qwen2.5VL-72B | 70.2 | 87.4 | 88.6 | 75.7 | 70.8 | 55.2 | 87.3 | 89.5 | 81.4 | 78.6 |
| InternVL3-9B | 57.7 | 84.7 | 83.4 | 70.5 | 66.3 | 51.2 | 79.6 | 86.2 | 75.5 | 41.2 |
| InternVL3-14B | 67.1 | 88.5 | 85.6 | 70.7 | 68.8 | 55.1 | 83.6 | 87.3 | 78.8 | 55.0 |
| InternVL3-38B | 70.1 | 90.1 | 87.6 | 75.6 | 71.5 | 57.1 | 85.0 | 89.2 | 81.3 | 62.4 |
| InternVL3-78B | 72.2 | 91.1 | 89.0 | 78.0 | 72.5 | 59.1 | 86.5 | 89.7 | 82.7 | 81.5 |
| OneVision-7B | 47.9 | 71.2 | 83.2 | 66.3 | 61.7 | 31.6 | 68.8 | 80.0 | 68.4 | 26.3 |
| LLaVA-OneVision-72B | 55.7 | 80.8 | 85.8 | 71.9 | 65.8 | 49.0 | 74.9 | 83.7 | 74.1 | 30.1 |
| Qwen2VL-7B | 54.1 | 83.1 | 83.0 | 70.1 | 60.7 | 50.6 | 76.5 | 83.0 | 72.9 | 16.3 |
| LLaVA-Next-7B | 37.6 | 63.2 | 69.2 | 57.8 | 37.6 | 27.6 | 31.6 | 51.9 | 49.8 | 8.5 |
| LLaVA-Next-13B | 37.3 | 62.3 | 70.0 | 57.6 | 40.4 | 31.8 | 34.9 | 59.0 | 51.6 | 9.7 |
| LLaVA-1.5-7B | 35.7 | 64.6 | 69.2 | 54.8 | 33.1 | 27.6 | 22.4 | 17.8 | 42.5 | 13.4 |
| LLaVA-1.5-13B | 37.0 | 63.6 | 66.5 | 55.3 | 34.3 | 24.5 | 24.9 | 18.5 | 42.9 | 15.0 |

*Table 6.* Performance comparison across benchmarks for different models measured by accuracy (%) and Vision Ratiio score (%).

$S_{\text{text}}$, text-based accuracy.

**The details of PCA Analysis** In Section 4.4, we use the PCA analysis regarding the Modality Preference Direction in Representation Space. Here, we provide a more detailed description of the setup. We extract the model's hidden representations from the last token of the input across different layers. Then, we apply the PCA method to reduce the dimensionality to two dimensions for visualization. The following settings were visualized:

1. The model states under the original modality context input in conflicting scenarios.

2. The model states when there is image noise or textual syntax errors.

3. The model states when specific instructions biased towards image or text are added.

To improve PCA dimensionality reduction efficiency, we selected 500 samples for each setting. Additionally, we calculated the center position after dimensionality reduction for each setting. The center (or centroid) of the samples is computed by taking the mean of the reduced-dimensional points across all the samples.

## D. Method Applying

### D.1. Details for Pattern of hidden states

In the main text, we visualize the layer-wise absolute difference and standard deviation of the hidden states for Qwen2.5VL-7B. As shown in Figure 11, we present the visualization of hidden states for OneVision-7B, Qwen2VL-7B, and InternVL3-8B. For each model, we selected layers with large absolute differences and small standard deviations. This means we identified the layers that showed stable and significant differences between instructions with modality preference towards vision context and text context, which are then used to steer and adjust the model's modality preference.

To provide a more rigorous explanation, we track the evolution of modality preference within the residual stream using a mechanistic logit-lens analysis. Specifically, under modality-conflict prompts, we project the layer-wise hidden states of the last token into the unembedding matrix of Qwen2.5-VL-7B, and calculate the Modality Preference Margin (MPM), defined as the average logit difference between the preferred modality and the competing modality. As shown in Fig. 12, the MPM remains close to zero in early layers, suggesting that these layers mainly focus on raw multimodal feature extraction. The modality preference first becomes distinguishable around Layers 20–23, indicating that these layers serve as the critical decision stage. Although later layers exhibit larger preference margins, they mainly reflect an already-formed decision. Therefore, Layers 20–23 constitute the most effective intervention point for steering modality preference, which is consistent with our empirical findings. We will incorporate these mechanistic analyses into the revised version.

### D.2. Evaluation of visual understanding and multi-modal multi-modal machine translation

PhD (Liu et al., 2024c) is a visual understanding benchmark and includes two subsets—PhD-ica, which contains irrelevant textual context, and PhD-icc, which introduces misleading or incorrect textual information—both of which increase the

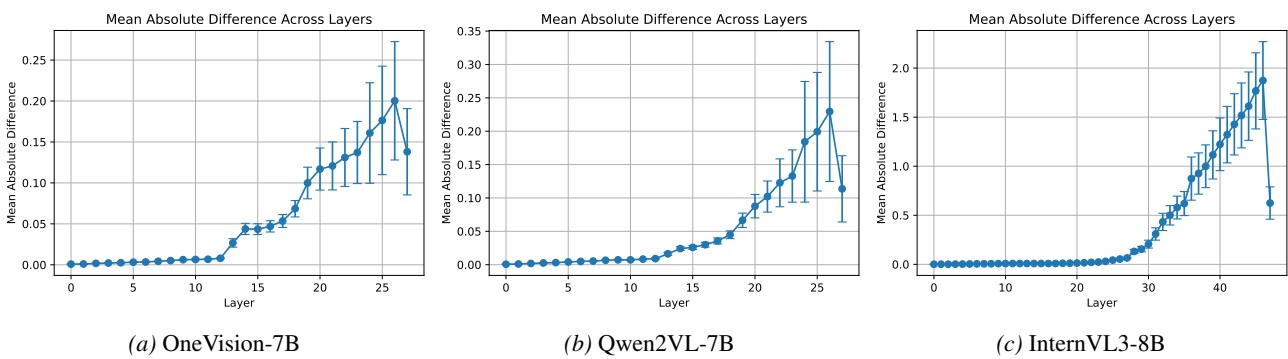

*(a)* OneVision-7B      *(b)* Qwen2VL-7B      *(c)* InternVL3-8B

*Figure 11.* Layer-wise absolute difference and standard deviation of hidden states between image-guided and text-guided instruction for OneVision-7B, Qwen2VL-7B and InternVL3-8B models from left to right.

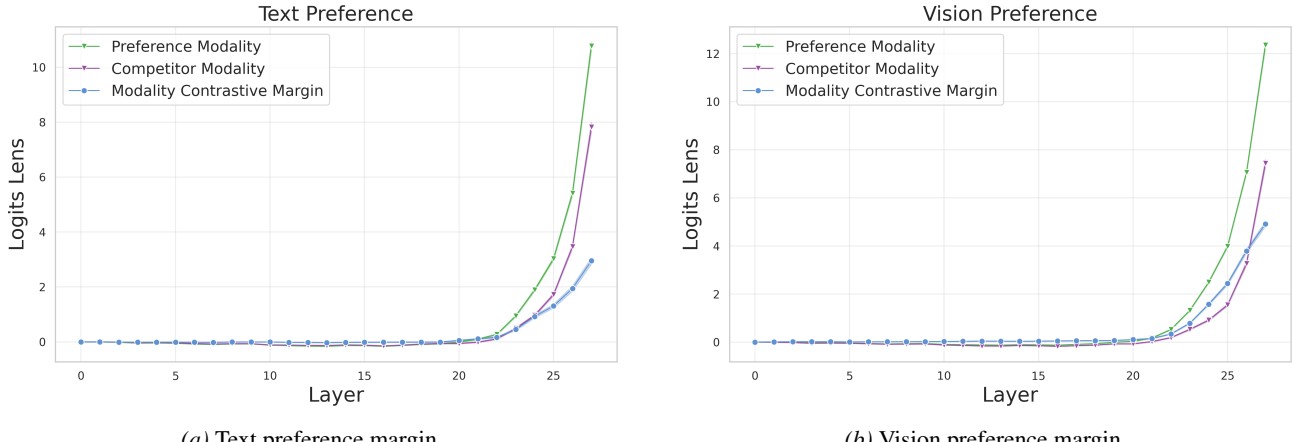

*(a)* Text preference margin.      *(b)* Vision preference margin.

*Figure 12.* Mechanistic logit-lens analysis of modality preference evolution across layers in Qwen2.5-VL-7B under modality-conflict prompts. The Modality Preference Margin (MPM) remains close to zero in early layers, first emerges around Layers 20–23, and is further amplified in later layers, suggesting that Layers 20–23 form the critical arbitration stage for modality-preference steering.

risk of hallucination. For testing convenience, we randomly selected 1,000 samples from the original Phd-cc and Phd-ica datasets for evaluation. By steering the model's modality preference toward the vision modality, we strengthen its visual understanding ability and mitigate vision hallucinations in MLLMs.

Ambigcaps (Li et al., 2021) benchmark explores the role of datasets in stimulating the leverage of the visual modality and proposes methods to highlight the importance of visual signals in the datasets. We evaluate the multi-modal machine translation (MMT) task on this dataset using the Qwen2.5VL-7B model. multi-modal contexts in MMT are both complementary and contradictory: the visual information provides helpful context for translation, but the potential for conflicting, non-visual signals can interfere with grounding the source language. Consequently, the proposed method is designed to steer the modality preference toward the text modality to ensure robustness against these visual-textual conflicts.

Conversely, when guided toward the text modality, the model places greater emphasis on the source language, leading to more accurate grounding in multi-modal machine translation. This adjustment prevents the model from over-relying on visual content and from introducing spurious objects or extraneous details into the translation output.

## E. More Experiment Analysis

In this section, we provide more experiment analysis including the sensitivity analysis for the evaluation of modality preference in Apdx E.1, the impact of text length on the evaluation of modality preference in Apdx E.2, the generation quality analysis of the proposed method in Apdx E.4, more results for downstream tasks in Apdx E.3, more detail for ablation study in Apdx E.5 and the further experiment without modality preference prior in Apdx E.6.

**E.1. The Sensitivity Analysis for the Evaluation of Modality Preference**

To verify whether the current **2k-sample scale** of $MC^2$ is sufficient to ensure the stability of both the preference evaluation and steering results, we conduct a sensitivity analysis by randomly selecting a specified quantity of samples from each category for assessment. As shown in Table 7, we calculate the Vision Ratio for OneVision-7B and Qwen2.5VL-7B. Results demonstrate that as the sample size increases per category, the Vision Ratio begins to stabilize around 150 samples. These experiments suggest that the current dataset size for each task is sufficient to ensure the stability of evaluation of modality preference. In the future, we would like to expand MC2 with a wider variety of tasks (e.g., texture recognition) and modalities (e.g., Audio), further enhancing its comprehensiveness and generalizability.

*Table 7.* The sensitivity analysis for the evaluation of modality preference. We randomly select a specified quantity of samples from each category for assessment, measured by Vision Ratio.

| Model | 25 | 50 | 75 | 100 | 125 | 150 | 175 | 200 | 225 | 250 |
|---|---|---|---|---|---|---|---|---|---|---|
| OneVision-7B | 29.2 | 28.0 | 29.1 | **29.6** | 28.1 | 26.4 | 26.6 | 26.7 | 26.7 | 26.3 |
| Qwen2.5VL-7B | 56.7 | 59.4 | 60.2 | **59.9** | 60.4 | 59.8 | 60.2 | 59.7 | 59.8 | 59.6 |

**E.2. The Impact of Text Length on the Evaluation of Modality Preference**

To investigate whether textual token length affects modality preference, we conduct an additional sensitivity analysis by adjusting the text length to approximately $0.6\times$–$1.3\times$ of the original length while preserving the core semantics. We then evaluate the Vision Ratio of Qwen2.5VL-7B, InternVL3-14B, and OneVision-7B under different text-length settings. As shown in Table 8, varying the textual length does not lead to significant shifts in modality preference. Across different length settings, the Vision Ratio remains relatively stable for all evaluated models. These results suggest that the modality preference measured by $MC^2$ is not substantially affected by moderate variations in textual length, further supporting the robustness of our evaluation.

*Table 8.* The sensitivity analysis for the impact of textual length on modality preference. We adjust the text length to approximately $0.6\times$–$1.3\times$ of the original length while preserving the core semantics, and measure the Vision Ratio under different length settings.

| Model | 0.6 | 0.8 | 1.0 | 1.2 | 1.3 |
|---|---|---|---|---|---|
| Qwen2.5VL-7B | 60.4 | 60.2 | 59.6 | 59.4 | 59.3 |
| InternVL3-14B | 56.9 | 54.6 | 55.0 | 53.8 | 54.2 |
| OneVision-7B | 25.7 | 25.5 | 26.3 | 25.2 | 25.3 |

**E.3. More Results for Downstream Tasks**

We evaluate the performance of Phd dataset using the proposed method for LLaVA-1.6-7B and the results are in Table 9. We observe that LLaVA-1.6-7B, due to its **severe text preference**, only achieves an average ACC score of $0.7$ on the PHD-icc subset. Applying our method significantly boosts the model's performance across the two subsets, thereby demonstrating the cross-model generalization of our approach.

**E.4. Generation Quality Analysis**

To further examine whether our method affects the quality of free-form text generation, we evaluate generation coherence and fluency on the AmbigCaps translation task with over 1000 samples. We measure coherence using perplexity (PPL, lower is better) and fluency using the GPT-win rate (higher is better) in both translation directions, i.e., English-to-Turkish (en→tr) and Turkish-to-English (tr→en). For the MLLM baseline, the PPL scores are $33.8/54.4$ for en→tr and tr→en, respectively, while the GPT-win rates are $45.8/53.4$. In comparison, our method achieves PPL scores of $24.2/60.8$ and GPT-win rates of $54.2/46.6$ in the two directions. These results show that our method improves generation coherence and fluency in the en→tr direction, while maintaining comparable quality in the tr→en direction. Overall, our method preserves competitive generation quality without causing catastrophic degradation in free-form text generation.

*Table 9.* Performance of the proposed method on the visual understanding benchmark, Phd using LLaVA-1.6-7B.

| Phd | Method | Attribute | Sentiment | Positional | Counting | Object | Avg |
|---|---|---|---|---|---|---|---|
| **Phd-icc** | LLaVA-1.6-7B | 0.5 | 0.0 | 0.0 | 1.5 | 1.5 | 0.7 |
| | InstDesign | 2.0 | 0.5 | 0.5 | 1.5 | 9.5 | 2.8 |
| | **Ours** | **3.5** | **1.5** | **1.0** | **3.0** | **15.0** | **4.8** |
| **Phd-iac** | LLaVA-1.6-7B | 5.5 | 8.0 | 4.0 | 10.5 | 29.0 | 11.4 |
| | InstDesign | 7.5 | 12.5 | 14.5 | 15.5 | 44.5 | 18.9 |
| | **Ours** | **11.5** | **20.0** | **20.5** | **22.0** | **51.0** | **25.0** |

## E.5. More Detail for Ablation Study

In this section, we conduct the detailed ablation study to analyze the proposed method.

### E.5.1. THE NUMBER OF PROBING SAMPLES

We compute the preference direction in Equation 1 using varied sample sizes but test the steering performance on the complete $MC^2$ dataset. We report the $S_{\text{Vision}}$ and $S_{\text{Text}}$ results for OneVision-7B and Qwen2.5VL-7B in Table 10. We observe that steering performance remains stable even when the steering vector is derived from a limited number of samples.

*Table 10.* The ablation study for the varied sample number of computing the preference direction.

| Model | 25 | 50 | 75 | 100 | 125 | 150 | 175 | 200 | 225 | 250 |
|---|---|---|---|---|---|---|---|---|---|---|
| OneVision-7B | 56.0 | 55.9 | 56.3 | **57.1** | 57.0 | 57.3 | 57.3 | 57.4 | 57.2 | 57.1 |
| Qwen2.5VL-7B | 62.2 | 62.2 | 61.7 | **62.6** | 64.0 | 62.8 | 62.7 | 63.1 | 62.8 | 63.6 |

### E.5.2. THE DIVERSITY OF PROBING SAMPLES

To study the impact of data diversity for probing task, we experiment by using **only a single task for probing** and applying the resulting vector to steer **all other tasks**. We report the $S_{\text{Vision}}$ and $S_{\text{Text}}$ for OneVision-7B and Qwen2.5VL-7B for entire dataset in Table 11. We observe that OneVision-7B achieves competitive performance compared to our initial implementation for using nearly each probing task, with Qwen2.5VL-7B showing similar success on over half the tasks. Further analysis finds that the most effective single-probing tasks are those where the initial modality preference change was **more pronounced**.

*Table 11.* The ablation study for the varied diversities of computing the preference direction.

| Model | Sport | Attribute | Sentiment | Positional | Counting | Color | Activity | Object | Ours |
|---|---|---|---|---|---|---|---|---|---|
| OneVision-7B | 58.6 | 55.8 | 59.3 | 57.8 | 59.0 | 53.4 | 58.5 | 58.6 | 57.1 |
| Qwen2.5VL-7B | 64.1 | 59.5 | 70.0 | 51.9 | 47.4 | 65.5 | 63.0 | 60.6 | 63.6 |

### E.5.3. DIFFERENT STEERING INTENSITIES

To investigate the performance with varied steering intensities, we introduce a scaling coefficient $\lambda$ to the steering weight $w$ in Equation 2 to change the steering intensity and conduct a test on $MC^2$ for both OneVision-7B and Qwen2.5-VL-7B, reporting the $S_{\text{Vision}}$ or $S_{\text{Text}}$ scores in Table 12. We observe that performance drops for both models with decreased steering intensity, indicating **insufficient steering**. As intensity increases, the performance of OneVision-7B significantly drops, exhibiting clear **over-steering** at $\lambda = 2.0$ which leads to destruction of language capabilities. Conversely, for Qwen2.5VL-7B, the steering effect continues to enhance up to $\lambda = 1.75$, only significantly degrading beyond $\lambda = 2.0$, demonstrating a wider **safe steering margin** in its representation space.

*Table 12.* The ablation study for the varied steering intensities.

| Model | 0.125 | 0.25 | 0.5 | 0.75 | 1.0 | 1.25 | 1.5 | 1.75 | 2.0 | 2.25 | 2.5 | 2.75 | 3.0 |
|---|---|---|---|---|---|---|---|---|---|---|---|---|---|
| OneVision-7B | 40.2 | 45.5 | 51.8 | 52.6 | 57.1 | 56.5 | 32.6 | 12.1 | 3.9 | 0.1 | 0.0 | 0.0 | 0.0 |
| Qwen2.5VL-7B | 40.8 | 44.1 | 49.4 | 53.2 | 63.6 | 72.3 | 76.0 | 70.7 | 45.8 | 15.2 | 16.2 | 12.5 | 7.8 |

### E.5.4. DETAILS OF IN-DEPTH ANALYSIS FOR STEERING METHOD

We provide the detailed description of the case for in-depth analysis for steering method in Figure 6 in Section 5.4. Besides, we also provide more attention analysis for the case in the different layers in Figure 14, 15, 16, 17, 18 and 19. We observe that across all subsequent layers following steering modality preference towards text at the 21 th layer, our method significantly increases the model's attention weight toward the text modality, surpassing the corresponding vision attention weight. This change clearly demonstrates that the steering mechanism successfully alters the modality preference by enhancing the model's dependency on the text modality.

### E.5.5. LATENCY AND MEMORY

*Table 13.* The comparison of inference time between MLLM-only and the proposed method including probing and steering phases.

| Dataset | MC2 | Phd-icc |
|---|---|---|
| MLLM-only | 1.99 | 1.84 |
| Probing (Offline) | 2.21 | 2.12 |
| Steering | 2.00 | 1.84 |

The proposed method consists of two phases, probing and steering. The probing phase is conducted offline and the resulting steering vector is cached for reuse. During the actual steering phase, we simply load this cached steering vector, which incurs minimal memory overhead and does not add meaningful computational cost to the inference process. We measure the single-sample inference latency (seconds) (without Flash-Attention acceleration and batch inference) for our method compared to the MLLM baseline (MLLM-only) in Table 13. The results show that the steering phase introduces negligible latency compared to the MLLM-only baseline, and the overhead is confined to the initial offline probing stage. Furthermore, all three methods require nearly identical memory requirements.

### E.5.6. THE PREREQUISITES FOR IMPLEMENTATION OF THE PROPOSED METHOD

Based on Representation Engineering (Greenblatt et al., 2023; Xu et al., 2024), the proposed method requires capturing an **explicit modality preference direction vector** to realize behavioral adjustment. The approach succeeds when such a vector can be reliably extracted, as seen in models like Qwen2.5VL-7B. However, the method fails in cases such as LLaVA-1.5-7B, primarily due to its limited ability to follow instructions for preference adjustment, which prevents the capture of a meaningful direction vector. Besides, we observe a localized performance drop in the Attribute subset of the Phd-icc benchmark in Table 3. We attribute this to the limitation of applying a single global steering vector, which may fail to accommodate **instance-level granularity**—where fine-grained, sample-specific features are required for optimal alignment. Despite these isolated cases, the overall benchmark performance improves, underscoring the effectiveness of our approach, especially considering that it requires no external data or fine-tuning.

### E.6. The Further Experiment without modality preference prior

Our current approach intentionally select the steering direction based on known task requirements. This design has been proven to be pragmatic and effective for many real-world applications where the optimal modality is clear. In addition, our method can be readily integrated with a training-free priority detection method to enable dynamic preference selection. To demonstrate this, we conduct the following experiment:

**Dataset Construction**: We modify $MC^2$ dataset by degrading the quality of one modality context so that only one modality is reliable, and the ground-truth answer aligns with it. We use QA accuracy to measure model performance on this new dataset.

**Task Design:** 1) Each sample requires a specific reliable modality. 2) All samples share the same reliable modality in a task. Each task contains 200 samples.

**Solution:** We apply a causal analysis approach (Parcalabescu & Frank, 2024) to identify the reliable modality. For each sample, we first measure the change of predicted answer probability when removing either the image or the text context. The larger the drop, the more important that modality is for the given sample. For Task1, we determine the reliable modality for a specific sample by comparing the probability drops. For Task2, by aggregating the reliable modalities across all samples via majority voting, we determine the preferred modality for the specific task.

**Results:** For the identification of reliable modality, we achieve an accuracy of $85.3\%$ for all samples in Task1; we reach $100\%$ accuracy for task-level identification in Task2 (thus, performance on Task 2 is equivalent to knowing the steering preference in advance). Next, we evaluate the performance of the proposed method on Task1, measured by QA accuracy in Table 14.

The results show that steering with predicted preference yields significant gains over base models and closely matches the performance of the "preference prior" setting. This confirms that our method can be simply adapted to autonomously prioritize modalities based on input quality or task needs.

*Table 14.* The performance of the proposed method on the revised $MC^2$ without modality preference prior measured by Accuracy.

| Method | Task1 |
|---|---|
| OneVision-only | 25.4 |
| +Steering with preference prior | 40.7 |
| +Steering with predicted preference | 37.5 |
| Qwen2.5VL-7B-only | 49.1 |
| +Steering with preference prior | 62.7 |
| +Steering with predicted preference | 58.2 |

Table 15. Accuracy of question answering in the MC$^2$ dataset when both textual and visual contexts are provided but the instruction does not specify which modality context should be used. Values are reported as vision-based accuracy/text-based accuracy for each model.

| Model | Sport | Attribute | Sentiment | Positional | Counting | Color | Activity | Object | Avg |
|---|---|---|---|---|---|---|---|---|---|
| LLaMAVision-11B | 31.2/52.4 | 20.4/69.6 | 2.0/93.2 | 21.2/66.8 | 4.0/93.2 | 35.2/47.2 | 10.0/82.4 | 38.8/42.8 | 20.4/68.4 |
| LLaVA1.5-7B | 20.0/59.6 | 8.0/88.0 | 2.0/86.0 | 8.8/75.2 | 1.2/96.0 | 10.8/82.0 | 9.6/78.8 | 35.2/52.0 | 11.9/77.2 |
| LLaVA1.5-13B | 34.4/59.6 | 8.8/89.6 | 4.8/88.0 | 12.0/84.8 | 1.6/96.4 | 12.8/82.0 | 9.6/87.6 | 31.2/62.8 | 14.4/81.3 |
| OneVision-7B | 32.0/36.4 | 21.6/54.8 | 2.8/94.4 | 24.8/56.4 | 2.4/86.4 | 30.0/38.0 | 11.6/71.2 | 42.4/31.2 | 20.9/58.6 |
| Owl3-24-07 | 60.8/31.6 | 16.4/72.4 | 10.8/85.6 | 22.0/69.6 | 8.4/88.0 | 28.4/60.4 | 17.2/71.2 | 60.0/29.6 | 28.0/63.5 |
| Qwen2VL-7B | 26.4/58.0 | 12.4/82.8 | 0.8/95.6 | 13.2/80.4 | 4.0/93.6 | 16.0/78.8 | 11.6/83.6 | 38.0/54.0 | 15.3/78.3 |
| Qwen2.5VL-7B | 65.6/12.8 | 45.2/46.0 | 18.0/68.8 | 46.4/38.0 | 51.6/39.6 | 70.8/20.0 | 42.0/43.6 | 77.6/14.0 | 52.2/35.4 |
| GLM-4V-9B | 42.0/42.4 | 32.4/59.2 | 8.8/81.6 | 28.0/62.4 | 15.2/74.4 | 56.8/32.8 | 23.3/66.0 | 54.0/32.8 | 32.6/56.5 |
| SPHINX-V2-1K | 39.6/50.8 | 14.8/82.4 | 1.2/98.4 | 16.8/77.6 | 9.2/85.6 | 23.2/69.2 | 24.4/67.2 | 59.2/32.4 | 23.6/70.5 |
| InternVL3-9B | 45.2/35.2 | 21.2/68.0 | 20.8/62.4 | 27.2/54.4 | 23.2/50.4 | 38.0/40.4 | 19.6/63.2 | 76.8/14.8 | 34.0/48.6 |
| InternVL3-14B | 72.8/8.8 | 30.8/48.4 | 25.2/60.0 | 33.2/52.0 | 37.2/47.2 | 58.0/21.2 | 24.8/52.8 | 84.4/9.6 | 45.8/37.5 |
| CogVLM2-19B | 44.0/39.6 | 29.2/56.0 | 8.8/75.6 | 19.2/54.8 | 8.0/73.2 | 31.6/43.2 | 25.2/60.8 | 59.2/28.4 | 28.2/54.0 |
| InternVL3-38B | 75.2/9.6 | 45.2/33.6 | 19.6/60.8 | 44.0/42.0 | 41.6/40.0 | 48.4/29.6 | 50.4/23.2 | 84.4/8.0 | 51.1/30.8 |
| InternVL3-78B | 92.4/3.2 | 46.0/28.8 | 66.4/18.4 | 41.6/37.2 | 69.6/13.2 | 76.4/8.8 | 74.4/12.8 | 89.6/4.0 | 69.5/15.8 |
| Qwen2.5-VL-32B | 85.60/10.40 | 49.20/39.20 | 49.60/42.80 | 52/37.60 | 52/42 | 70.80/20 | 57.20/35.20 | 86.80/10.40 | 62.90/29.70 |
| Qwen2.5VL-72B | 93.6/4.4 | 59.2/27.2 | 50.0/41.2 | 73.6/19.2 | 63.6/29.2 | 83.6/9.6 | 74.0/21.2 | 89.2/8.0 | 73.4/20.0 |
| OneVision-72B | 47.2/46.0 | 20.0/70.8 | 4.0/93.6 | 22.8/67.2 | 12.8/83.2 | 21.6/60.8 | 20.8/70.8 | 71.6/21.2 | 27.6/64.2 |
| LLaVA1.6-7B | 10.8/74.4 | 5.2/85.2 | 0.8/93.2 | 3.6/79.6 | 0.4/90.8 | 6.0/76.0 | 4.8/73.6 | 26.0/46.8 | 7.2/77.5 |
| LLaVA1.6-13B | 16.0/66.4 | 7.2/90.4 | 0.8/92.0 | 6.4/91.6 | 2.4/95.6 | 6.8/88.0 | 10.0/84.4 | 22.4/63.2 | 9.0/84.0 |
| LLaVA1.6-34B | 34.8/42.4 | 12.0/81.6 | 6.8/85.6 | 16.8/76.0 | 11.2/83.2 | 25.2/60.8 | 14.0/76.0 | 60.0/31.6 | 22.6/67.2 |
| GPT-4o-mini | 94.4/3.2 | 35.6/47.6 | 60.4/28.4 | 22.0/58.9 | 19.4/59.2 | 19.4/36.4 | 71.2/20.4 | 78.4/12.8 | 52.0/33.4 |

Table 16. Accuracy of question answering in the MC$^2$ dataset when both textual and visual contexts are provided and the instruction explicitly directs the model to answer based on visual modality context. Values are reported as vision-based accuracy/text-based accuracy for each model.

| Model | Sport | Attribute | Sentiment | Positional | Counting | Color | Activity | Object | Avg |
|---|---|---|---|---|---|---|---|---|---|
| OneVision-7B | 55.6/16.4 | 31.2/37.2 | 12.0/76.8 | 30.8/42.4 | 3.2/77.6 | 36.4/18.4 | 22.4/47.6 | 61.2/16.4 | 31.6/41.6 |
| Qwen2VL-7B | 60.8/26.8 | 24.0/69.2 | 20.0/69.6 | 20.4/74.0 | 10.8/80.0 | 32.0/52.0 | 27.2/61.6 | 63.2/28.8 | 32.3/57.8 |
| Qwen2.5VL-7B | 77.6/14.4 | 43.2/46.8 | 18.4/72.8 | 43.2/40.4 | 35.6/55.6 | 58.8/24.4 | 53.6/35.6 | 81.2/11.6 | 51.4/37.7 |
| CogVLM2-19B | 73.2/13.2 | 47.6/32.4 | 35.6/28.4 | 26.8/45.2 | 14.0/40.0 | 61.6/17.6 | 56.0/28.0 | 76.0/15.2 | 48.9/27.5 |
| InternLM-XC2.5-7B | 84.0/9.6 | 46.4/42.8 | 74.0/18.4 | 36.0/52.4 | 22.8/66.0 | 63.6/20.8 | 74.0/18.4 | 76.4/15.6 | 59.7/30.5 |
| GLM-4V-9B | 75.2/18.4 | 48.8/39.6 | 28.8/54.0 | 33.6/55.6 | 38.4/54.0 | 76.4/16.4 | 48.4/38.8 | 80.0/11.6 | 53.7/36.1 |
| SPHINX-V2-1K | 52.4/38.4 | 16.4/78.8 | 2.0/97.2 | 20.8/72.8 | 13.6/80.8 | 30.0/58.8 | 40.8/52.8 | 64.8/29.2 | 30.1/63.6 |
| InternVL3-9B | 96.0/2.0 | 67.2/18.8 | 82.8/13.2 | 54.8/26.4 | 55.6/21.2 | 84.4/7.6 | 82.8/6.4 | 91.6/4.0 | 76.9/12.4 |
| InternVL3-14B | 98.4/0.8 | 86.0/4.4 | 87.6/7.6 | 71.6/12.8 | 78.0/6.8 | 97.2/0.8 | 90.8/3.2 | 96.4/1.6 | 88.2/4.8 |
| LLaVA1.6-7B | 33.2/54.0 | 6.8/80.8 | 6.0/82.4 | 6.4/79.6 | 2.8/90.4 | 10.8/70.0 | 13.6/69.2 | 48.4/40.8 | 16.0/70.9 |
| LLaVA1.6-13B | 41.6/40.4 | 10.4/85.2 | 4.0/62.8 | 8.4/83.6 | 5.6/92.8 | 14.4/70.8 | 24.8/58.0 | 45.2/41.2 | 19.3/66.9 |
| LLaVA1.6-34B | 84.8/12.0 | 48.0/36.4 | 62.8/24.0 | 34.0/52.0 | 38.8/44.4 | 76.4/14.4 | 62.0/18.4 | 80.4/12.4 | 60.9/26.8 |

**<image> represents the visual input displayed below**

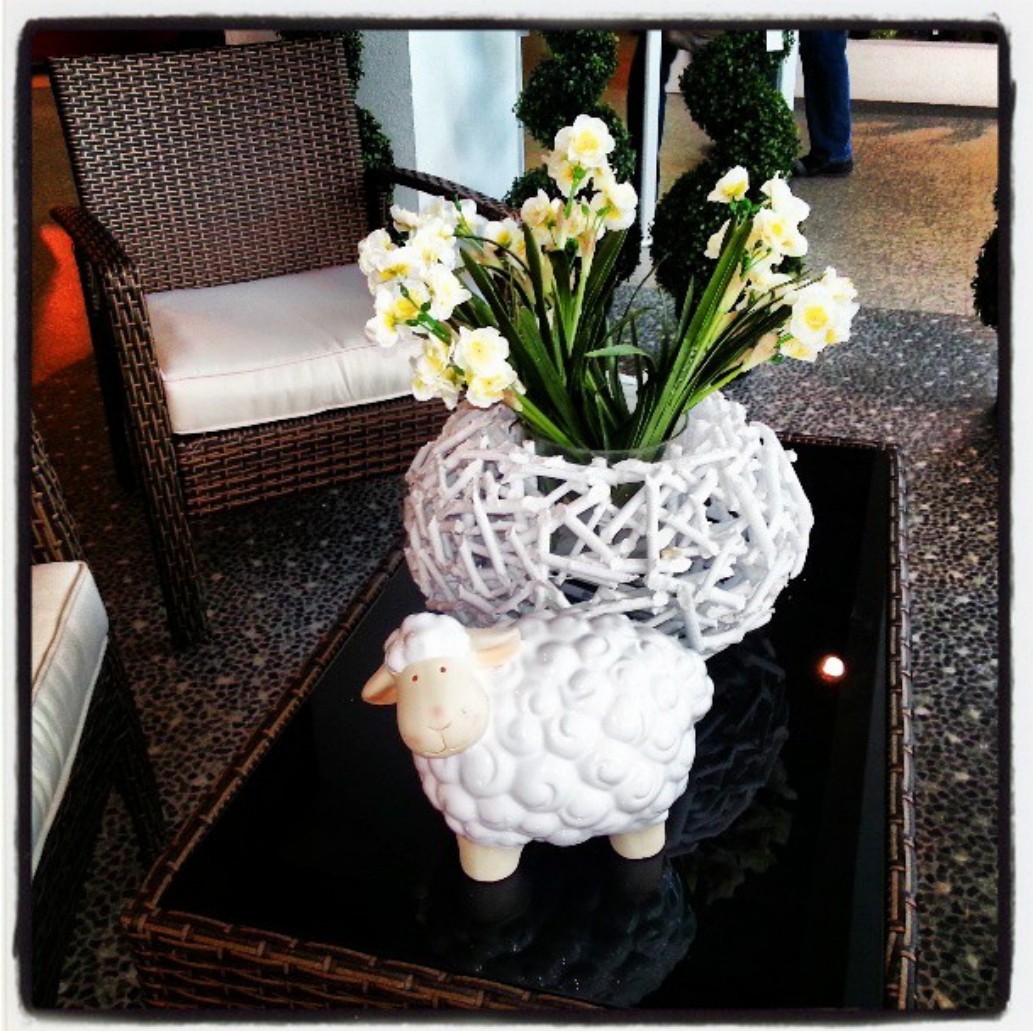

**User:** ⟨image⟩ **Text Context:** The table was adorned with a vibrant bouquet of flowers and a charming ceramic sheep, while the surrounding chairs, crafted from smooth, *polished wood*, complemented the rustic yet elegant setting. In case of inconsistency, prioritize the text context.
**Question:** What **are** the chairs made of? A. wicker     B. wood
**Assistant:** ⟨output⟩

**Vision-based answer:** *wicker*
**Text-based answer:** *wood*
**Baseline answer:** A. wicker ✗
**Proposed steering answer:** B. wood ✓

*Figure 13.* Example of a cross-modal conflict sample from our diagnostic dataset. The highlighted text indicates the ground-truth modality specified by the instruction, which the proposed steering method successfully follows while the baseline fails.

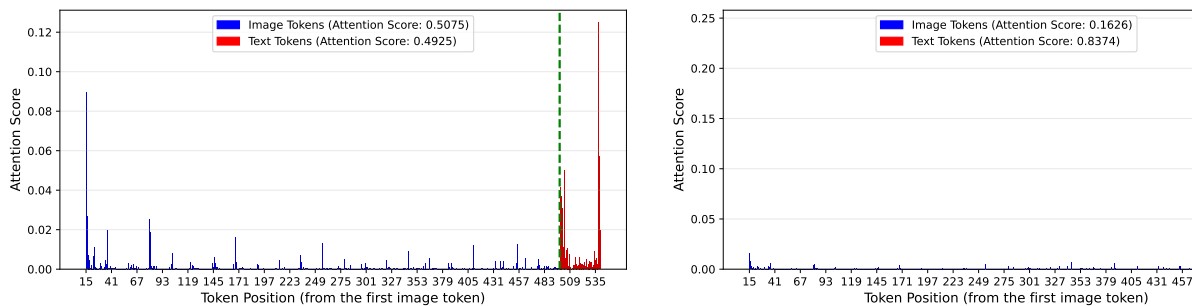

*Figure 14.* The attention scores of the generated token toward the vision and text contexts using InstDesign (Left) and the proposed method (Right) in the 23rd layer.

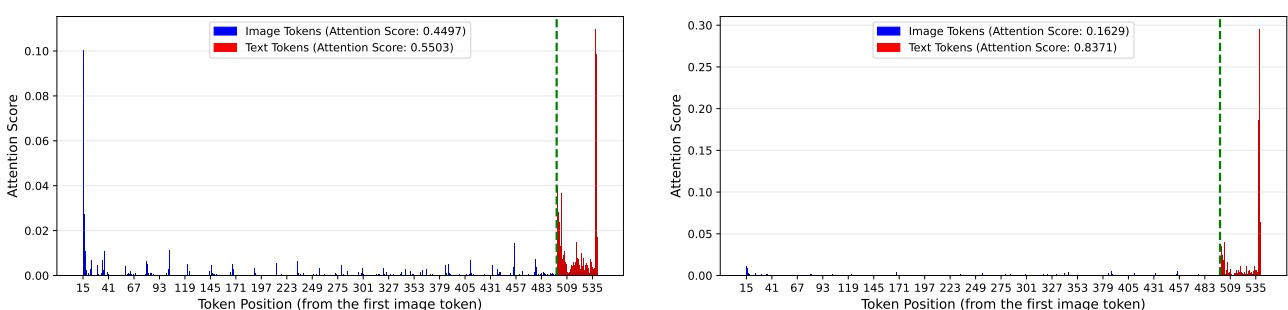

*Figure 15.* The attention scores of the generated token toward the vision and text contexts using InstDesign (Left) and the proposed method (Right) in the 24th layer.

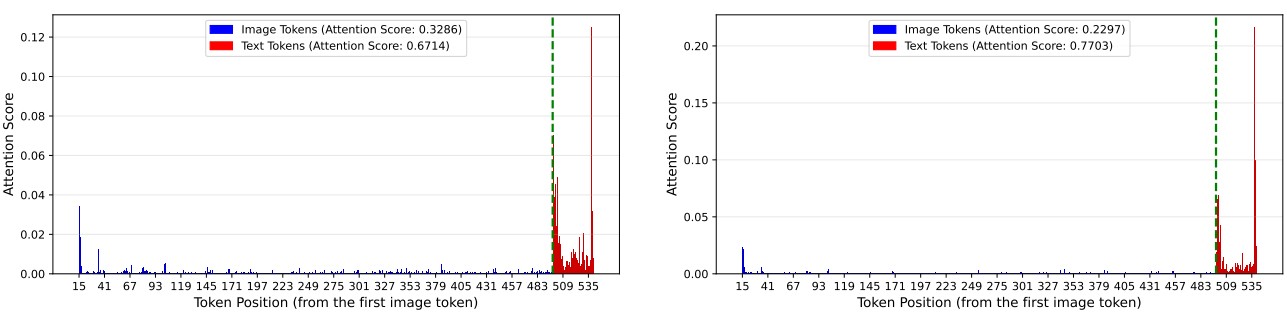

*Figure 16.* The attention scores of the generated token toward the vision and text contexts using InstDesign (Left) and the proposed method (Right) in the 25th layer.

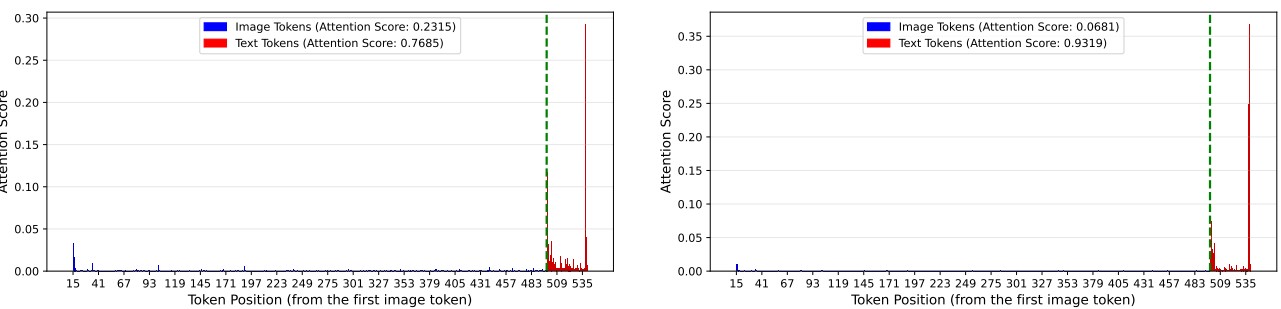

*Figure 17.* The attention scores of the generated token toward the vision and text contexts using InstDesign (Left) and the proposed method (Right) in the 26th layer.

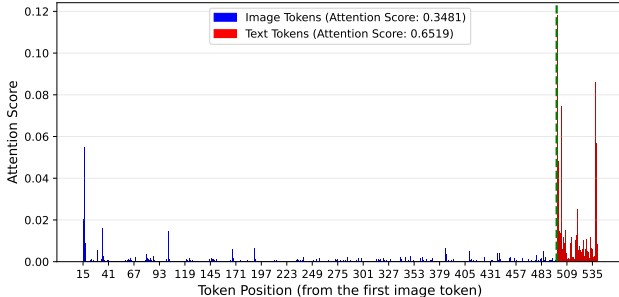 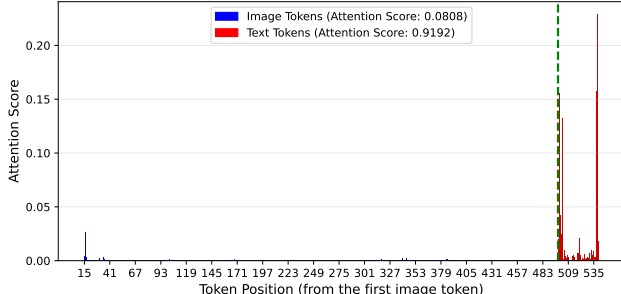

*Figure 18.* The attention scores of the generated token toward the vision and text contexts using InstDesign (Left) and the proposed method (Right) in the 27th layer.

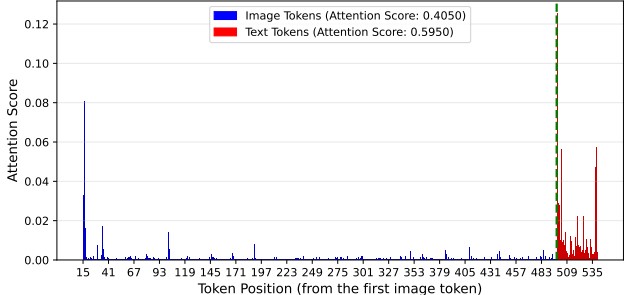 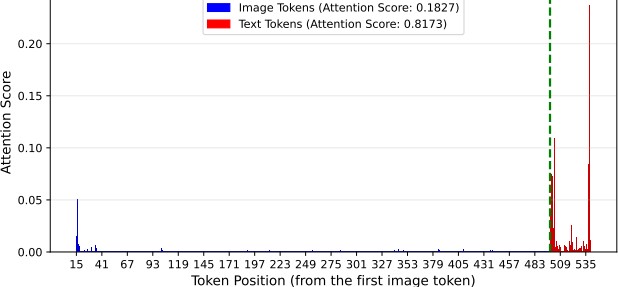

*Figure 19.* The attention scores of the generated token toward the vision and text contexts using InstDesign (Left) and the proposed method (Right) in the 28th layer.

*Table 17.* Accuracy of question answering in the MC$^2$ dataset when both textual and visual contexts are provided and the instruction explicitly directs the model to answer based on the textual modality. Values are reported as vision-based accuracy/text-based accuracy for each model.

| Model | Sport | Attribute | Sentiment | Positional | Counting | Color | Activity | Object | Avg |
|---|---|---|---|---|---|---|---|---|---|
| **OneVision-7B** | 45.6/16.4 | 22.4/37.2 | 5.6/76.8 | 27.2/42.4 | 1.6/77.6 | 28.8/18.4 | 16.8/47.6 | 56.0/16.4 | 25.5/41.6 |
| **Qwen2VL-7B** | 51.6/34.8 | 14.8/78.4 | 6.8/88.0 | 15.6/79.6 | 4.0/90.8 | 18.0/70.8 | 19.2/72.8 | 57.6/36.0 | 23.4/68.9 |
| **Qwen2.5VL-7B** | 77.6/14.4 | 43.2/46.8 | 18.4/72.8 | 43.2/40.4 | 35.6/55.6 | 58.8/24.4 | 53.6/35.6 | 81.2/11.6 | 51.4/37.7 |
| **CogVLM2-19B** | 53.6/29.6 | 28.4/47.6 | 10.4/56.8 | 17.6/56.8 | 6.0/62.0 | 36.0/35.2 | 34.0/39.6 | 67.2/19.6 | 31.6/43.4 |
| **GLM-4V-9B** | 53.2/32.8 | 30.4/61.2 | 6.0/85.6 | 23.2/68.0 | 20.0/70.0 | 52.4/35.6 | 28.0/60.8 | 68.0/22.0 | 35.2/54.5 |
| **SPHINX-V2-1K** | 48.4/41.2 | 14.4/81.6 | 2.0/98.0 | 19.2/77.6 | 11.2/84.0 | 27.2/67.2 | 30.8/65.2 | 63.2/30.8 | 27.1/68.2 |
| **InternVL3-9B** | 41.2/27.2 | 13.6/71.6 | 22.4/60.8 | 16.8/64.0 | 18.0/60.4 | 25.6/46.4 | 29.2/49.2 | 62.4/17.6 | 28.6/49.6 |
| **InternVL3-14B** | 28.4/44.8 | 14.0/68.4 | 3.6/82.4 | 21.2/54.8 | 28.0/43.2 | 24.8/50.0 | 17.2/58.8 | 55.2/19.6 | 24.0/52.8 |

Table 18. Accuracy of question answering in the MC$^2$ dataset when only unimodal textual context is provided.

| Model | Sport | Attribute | Sentiment | Positional | Counting | Color | Activity | Object | Avg |
|---|---|---|---|---|---|---|---|---|---|
| **LLaMAVision** | 97.6 | 97.2 | 99.6 | 99.2 | 97.2 | 96.0 | 97.6 | 97.6 | 97.8 |
| **LLaVA1.5-7B** | 98.0 | 98.0 | 100.0 | 97.6 | 98.4 | 99.2 | 97.6 | 97.6 | 98.3 |
| **LLaVA1.5-13B** | 97.2 | 97.6 | 99.6 | 97.6 | 97.6 | 98.8 | 95.2 | 99.2 | 97.9 |
| **OneVision-7B** | 98.0 | 95.2 | 100.0 | 98.4 | 98.0 | 98.8 | 98.0 | 100.0 | 98.3 |
| **Owl3** | 97.6 | 97.2 | 99.6 | 98.8 | 98.8 | 99.2 | 99.2 | 100.0 | 98.8 |
| **Qwen2VL-7B** | 98.8 | 96.4 | 99.6 | 99.6 | 98.8 | 100.0 | 98.8 | 100.0 | 99.0 |
| **Qwen2.5VL-7B** | 99.2 | 97.6 | 100.0 | 99.6 | 96.8 | 98.8 | 98.4 | 99.2 | 98.7 |
| **CogVLM2-19B** | 98.0 | 95.2 | 99.2 | 96.0 | 94.8 | 98.4 | 98.0 | 99.6 | 97.4 |
| **GLM-4V-9B** | 98.4 | 95.2 | 99.6 | 97.2 | 98.8 | 98.4 | 99.6 | 99.6 | 98.4 |
| **SPHINX-V2-1K** | 98.4 | 97.6 | 99.2 | 98.8 | 98.0 | 99.2 | 98.4 | 99.6 | 98.7 |
| **InternVL3-9B** | 97.6 | 98.0 | 99.6 | 99.2 | 95.6 | 96.8 | 98.8 | 99.2 | 98.1 |
| **InternVL3-14B** | 98.4 | 98.4 | 100.0 | 99.2 | 95.6 | 98.4 | 98.8 | 99.6 | 98.5 |
| **InternVL3-38B** | 97.6 | 96.8 | 100.0 | 98.8 | 96.0 | 97.2 | 98.4 | 100.0 | 98.1 |
| **InternVL3-78B** | 97.2 | 97.6 | 100.0 | 98.0 | 96.4 | 96.8 | 98.0 | 100.0 | 98.0 |
| **Qwen2.5VL-72B** | 99.6 | 98.4 | 96.8 | 100.0 | 97.2 | 100.0 | 99.6 | 99.2 | 98.9 |
| **OneVision-72B** | 100.0 | 97.6 | 97.6 | 99.6 | 96.4 | 100.0 | 100.0 | 98.8 | 98.7 |
| **GPT-4o-mini** | 97.6 | 97.2 | 99.6 | 98.6 | 97.4 | 98.4 | 98.4 | 100.0 | 98.4 |

Table 19. Accuracy of question answering in the MC$^2$ dataset when only unimodal visual context is provided.

| Model | Sport | Attribute | Sentiment | Positional | Counting | Color | Activity | Object | Avg |
|---|---|---|---|---|---|---|---|---|---|
| **LLaMAVision** | 100.0 | 98.8 | 92.8 | 98.4 | 96.4 | 99.2 | 98.8 | 97.2 | 97.7 |
| **LLaVA1.5-7B** | 99.6 | 98.0 | 96.4 | 100.0 | 97.6 | 99.6 | 98.8 | 98.4 | 98.5 |
| **LLaVA1.5-13B** | 99.6 | 95.2 | 94.4 | 97.6 | 95.2 | 98.4 | 96.4 | 98.4 | 96.9 |
| **OneVision-7B** | 100.0 | 97.2 | 97.2 | 98.4 | 84.4 | 99.6 | 97.2 | 98.8 | 96.6 |
| **Owl3** | 99.2 | 94.0 | 94.0 | 97.2 | 88.4 | 96.8 | 97.2 | 99.2 | 95.8 |
| **Qwen2VL-7B** | 99.6 | 98.8 | 95.6 | 98.4 | 96.4 | 100.0 | 99.6 | 98.4 | 98.3 |
| **Qwen2.5VL-7B** | 99.6 | 98.8 | 98.0 | 100.0 | 99.2 | 100.0 | 100.0 | 98.8 | 99.3 |
| **CogVLM2-19B** | 99.6 | 99.2 | 91.2 | 96.8 | 91.6 | 98.8 | 98.4 | 98.8 | 96.8 |
| **GLM-4V-9B** | 99.6 | 99.2 | 98.0 | 99.2 | 97.6 | 100.0 | 99.2 | 99.6 | 99.1 |
| **SPHINX-V2-1K** | 98.8 | 97.6 | 99.2 | 92.8 | 98.0 | 99.6 | 96.8 | 99.2 | 97.8 |
| **InternVL3-9B** | 98.8 | 95.6 | 95.6 | 96.8 | 90.0 | 100.0 | 98.0 | 98.0 | 96.6 |
| **InternVL3-14B** | 99.2 | 96.4 | 96.4 | 98.4 | 92.4 | 98.8 | 97.2 | 98.4 | 97.1 |
| **InternVL3-38B** | 100.0 | 98.0 | 97.2 | 100.0 | 94.4 | 99.6 | 99.2 | 98.8 | 98.4 |
| **InternVL3-78B** | 99.2 | 99.6 | 96.8 | 98.8 | 96.0 | 100.0 | 99.2 | 98.4 | 98.5 |
| **Qwen2.5VL-72B** | 97.2 | 97.2 | 100.0 | 99.2 | 97.2 | 98.4 | 98.4 | 99.6 | 98.4 |
| **OneVision-72B** | 100.0 | 97.6 | 97.6 | 99.6 | 96.4 | 100.0 | 100.0 | 98.8 | 98.7 |
| **GPT-4o-mini** | 100.0 | 92.0 | 95.6 | 100.0 | 100.0 | 96.0 | 96.4 | 96.0 | 97.0 |

