# OpenReview forum: "Evaluating and Steering Modality Preferences in Multi-modal LLMs"
_ICML.cc/2026/Conference — ICML 2026 regular_

### Official Review · Reviewer_abYr · 2026-03-05

**Soundness:** 2
**Presentation:** 3
**Significance:** 3
**Originality:** 3
**Overall Recommendation:** 4
**Confidence:** 4

**Summary:**

This paper studies **modality preference in multimodal large language models (MLLMs)**—whether models rely more on visual evidence or textual context when the two sources conflict. The authors introduce **MC2**, a benchmark that constructs image–text conflict cases to measure preference using a metric called *Vision Ratio*. Experiments across around twenty MLLMs suggest that many models exhibit strong textual bias and that modality preference correlates with downstream multimodal performance. The paper also proposes a **training-free steering method** based on representation probing to adjust modality preference at inference time.

**Compliance With Llm Reviewing Policy:**

Affirmed.

**Final Justification:**

The paper studies a relevant and under-explored question—modality preference in MLLMs—with a well-designed benchmark and useful steering method. The rebuttal was thorough: the manual re-annotation experiment convincingly mitigates benchmark–metric coupling concerns, reproducibility across LLM versions is demonstrated, and steering robustness is shown. I appreciate the authors' clarification that the Vision Ratio–performance relationship is correlational rather than causal, which appropriately scopes the claim. My residual concern is that the steering-based causal evidence still cannot fully disentangle modality preference from other representational effects. Nonetheless, the overall contribution—benchmark, analysis, and practical steering method—is solid and likely to be built upon. I maintain my score of 4.

**Key Questions For Authors:**

1. **Benchmark–metric coupling**

   How do the authors ensure that Vision Ratio reflects genuine modality preference rather than artifacts introduced by binary-choice reformulation or the linguistic properties of generated textual contexts?

2. **Dataset reproducibility**

   Will the authors release the finalized MC2 dataset along with the full generation pipeline (prompts, filtering rules, and sampling parameters)? How stable are the results if the dataset is regenerated using different LLM versions?

3. **Correlation vs. causation**

   Does Vision Ratio still predict downstream performance after controlling for model size, architecture family, or baseline task accuracy?

4. **Steering robustness**

   How sensitive is the steering method to the choice of layer, steering coefficient, or probing prompts used to extract the modality preference direction?

5. **External validity of MC2**

   Have the authors tested whether the conclusions about modality preference hold on datasets constructed without LLM-generated textual conflicts or in settings beyond binary-choice questions?

**Limitations:**

No.

The authors should explicitly discuss:

- the potential coupling between benchmark construction and the evaluation metric
- reproducibility limitations associated with LLM-generated benchmark data
- whether the correlation between Vision Ratio and downstream capability reflects causation or confounding factors

**Strengths And Weaknesses:**

### Strengths

1. **Clear and relevant research question.**
   The paper studies modality preference in MLLMs, an important but under-explored aspect of multimodal reasoning.

2. **Reasonable benchmark design with quality controls.**
   The MC2 dataset includes LLM-based generation, filtering, and human verification to ensure that visual and textual evidence independently support different answers.

3. **Empirical analysis across many models.**
   The paper evaluates a relatively large set of MLLMs and provides additional analyses such as attention patterns and preference-shift experiments.

4. **Practical steering approach.**
   The proposed training-free representation steering method is simple and lightweight, and shows promising improvements in several tasks.

---

### Weaknesses

1. **Potential benchmark–metric coupling.**
   The Vision Ratio metric is tightly tied to the MC2 construction process (binary-choice reformulation and LLM-generated conflicts), which may partially reflect dataset artifacts rather than general modality preference.

2. **Reproducibility concerns due to LLM-generated data.**
   Since MC2 relies on LLM-generated contexts and filtering, the dataset generation process may be sensitive to model versions, prompts, and sampling settings.

3. **Correlation results may be confounded.**
   The paper reports a strong correlation between Vision Ratio and downstream multimodal performance. However, this relationship may be driven by confounding factors such as model scale, architecture, or training data. Stronger models may both perform better on vision tasks and exhibit different modality preferences, making it unclear whether Vision Ratio itself provides explanatory or predictive value beyond overall model capability.

4. **Limited methodological novelty.**
   The steering method is conceptually similar to existing activation steering / representation editing techniques.

---

> ### Author Rebuttal · Authors · 2026-03-31
>
> >W1 & Q1: Benchmark design
>
> **R1:** Thanks for your comments. Evaluating modality preference directly in natural scenarios is confounded by several key factors and we isolate these factors by:
> 1. **Binary-choice reformulation** Constrains the output space to binary vision-text preference space to exclude the dispersion of the language output, irrelevant to modality preference. Besides, Vision Ratio explicitly mitigates choice position bias in Section 3.3.
> 2. **LLM-generated controllable conflicts** ensure unimodal content is precisely understood by MLLMs, decoupling modality preference from unimodal comprehension and internal knowledge. Besides, as specified by the prompt constraints in the Appendix, **the generated text maintains high fidelity to the original (non-LLM-generated) captions**. Further, we verify that vision ratio is relatively stable across different LLM versions in **R3**.
> 3. **Generalizability to Real-World Scenarios** in Section 4.3 and Section 5, provide strong evidence that Vision Ratio can reflect MLLMs' visual understanding capabilities and these findings generalize effectively to real-world benchmarks.
>
> We will explicitly discuss the evaluation limitations, design philosophy and its advantages in the revised version.
> >W2 & Q2: Reproducibility LLM-Generated Data
>
> **R2:** According to your suggestion, we test the vision ratio using the data generated by ChatGPT or DeepSeek to test the stability of Vision Ratio, as follows:
>
> |Model|Original|ChatGPT|DeepSeek|
> |:-|:-|:-|:-|
> |Qwen2.5VL-7B|59.6|60.5|58.4|
> |OneVision-7B|26.3|25.2|27.8|
> |InternVL3-8B|41.2|40.9|41.6|
>
> Results show that Vision Ratio of the data generated by both models are relatively stable. And we commit to releasing the finalized MC2 dataset along with the full generation pipeline.
> >W3 & Q3: Correlation between Vision Ratio and downstream capability.
>
> **R3:** To isolate confounding factors, we steer the same MLLM to yield different Vision Ratios and evaluate its downstream performances of several visual understanding tasks. The results are summarized as follows:
>
> |Vision Ratio(Qwen2.5VL-7B)|MMMU|HallBench|RealworldQA|
> |:-|:-|:-|:-|
> |57.6|56.2|49.3|67.2|
> |59.6(Original)|58.6|52.9|68.5|
> |60.7|58.9|53.2|68.8|
> |62.8|61.2|54.1|69.1|
>
> |Vision Ratio(OneVision-7B)|MMMU|HallBench|RealworldQA|
> |:-|:-|:-|:-|
> |24.6|43.6|25.2|59.3|
> |26.3(Original)|47.9|31.6|66.3|
> |27.4|48.1|33.2|67.2|
> |29.5|49.0|34.1|68.1|
>
> The results show that the downstream task performances improve as Vision Ratio increases, when these cofounding factors remain fixed.
> >W4: Methodological novelty.
>
> **R4:** Thanks for your comment. The proposed method is conceptually somewhat similar to existing methods. However, our method advances in two key ways:
>
> 1. Unlike token-wise steering in existing method, we apply a unified steering vector to vision and text tokens, which ensures relative distribution of representations between vision and text modalities remains unchanged, stabilizing the inherent capabilities of the model.
> 2. The proposed method can adaptively adjust the steering weight without the need for manual enumeration.
>
> Please refer to the **R1 for Reviewer gAqo** for more detailed explanations and empirical results. We will include these in the revised version.
> >Q4: Steering robustness analysis
>
> **R5:** To evaluate the robustness of the steering method, we conducted experiments across different layers (including the layers immediately preceding and succeeding the selected one in the paper), various steering coefficients, and different probing prompts. The results are as follows:
>
> |Model|ours|Layer-1| Layer+1|Prompt2|Prompt3|1.4|1.2|0.8|
> |:-|:-|:-|:-|:-|:-|:-|:-|:-|
> |Qwen2.5VL-7B|63.6|63.0|62.4|63.8|62.2|65.5|63.2|62.0|
> |OneVision-7B|57.1|59.0|58.4|57.1|57.4|57.6|57.4|56.1|
>
> We observe that the steering method is relatively robust to the choice of layer, steering coefficient, and probing prompts. And we will include these results in the revised version.
> >Q5: External validity beyond MC2
>
> **R6:** Following your suggestion, we extended our evaluation to an open-ended format and the results are as follows:
>
> |Model|Orignial|Open-ended|
> |:-|:-|:-|
> |Qwen2.5VL-7B|59.6|61.6|
> |OneVision-7B|26.3|27.1|
> |InternVL3-8B|41.3|40.5|
>
> The results show that open-ended assessment aligns with our original evaluation. We will test more MLLMs and include these results in the revised version.
>
> >Limitations
>
> **R7:** Thanks for your suggestions! Building upon our responses in R1, R2, and R3, we will explicitly elaborate on our benchmark’s design philosophy, its analytical scope. And we commit to explicitly discuss the potential limitations within the Limitation sections in the revised version.

---

> > ### Author Rebuttal · Reviewer_abYr · 2026-04-01
> >
> > The rebuttal is helpful and adds relevant new evidence, but my main concerns are only partially resolved: the benchmark–metric coupling issue is mitigated but not fully ruled out, and the evidence on the relationship between Vision Ratio and downstream performance still does not completely disentangle modality preference from other effects introduced by steering.
> >
> > Overall, the work remains technically solid and well-executed, and I will maintain my current score.

---

> > > ### Author Response · Authors · 2026-04-04
> > >
> > > Thank you for your response. We sincerely appreciate your recognition that the work is **technically solid and well-executed**. We highly value your feedback and have made additional efforts to further address the issues you raised.
> > >
> > > # *Q1: Additional evidence on benchmark–metric coupling from manual re-annotation of $MC^2$*
> > >
> > > To directly examine whether Vision Ratio depends on artifacts introduced by LLM-generated textual conflicts, we manually re-annotated the text contexts in $MC^2$ while keeping the images unchanged.
> > >
> > > ### **1. Rigorous manual annotation**
> > >
> > > For each original sample, the conflicting text context was independently rewritten by three annotators. Annotators were instructed to preserve the original scene semantics, style, and non-target details as much as possible, while only modifying the answer-relevant evidence so that the text supports the conflicting answer. The final text context was determined by majority agreement and then manually verified for **fluency, grammatical correctness, and semantic consistency with the image except for the intended conflict**.
> > >
> > > ### **2. Large-scale re-evaluation**
> > > As with the original $MC^2$, we first verified that the evaluated models still achieved near-perfect unimodal text understanding (**around 99% accuracy**) on the manually annotated version. We then re-evaluated a broad set of MLLMs on this revised benchmark.
> > >
> > > MLLMs | Original (%) | Manual-annotated (%) | Relative Change (%)|
> > > |:-|:-|:-|:-|
> > > |LLamaVision|23.0|21.5| -6.5|
> > > |LLaVA1.5-7B|13.4|13.2| -1.5|
> > > |OneVision-7B|26.3|24.5| -6.8|
> > > |Owl3-7B|30.6|30.1| -1.6|
> > > |Qwen2VL-7B|16.3|16.0| -1.8|
> > > |CogVLM2-19B|34.3|35.4| 3.2|
> > > |GLM-4V-9B|36.6|37.5| 2.5|
> > > |SPHINX-V2-1K|25.1|25.3| 0.8|
> > > |LLaVA-Next-7B|8.5|8.5| 0.0|
> > > |InternVL3-14B|55.0|56.1| 2.0|
> > > |Qwen2.5VL-7B|59.6|62.1| 4.2|
> > > |Qwen2.5VL-32B|67.9|69.5| 2.4|
> > > |InternVL3-38B|62.4|62.0| -0.6|
> > >
> > > ### **3. Consistent results**
> > > The results remain highly stable across models on the manually annotated version, with most changes within **5%**, and the overall trends and conclusions remain unchanged. This suggests that **Vision Ratio is not primarily driven by artifacts specific to the LLM-generated textual contexts**.
> > >
> > > We will release the full dataset, the detailed annotation process, and the code for reproducibility.
> > >
> > >
> > > # *Q2: Relationship Between Vision Ratio and Downstream Performance*
> > >
> > > We agree that the observed relationship between Vision Ratio and downstream multimodal performance may be influenced by confounding factors such as model scale, architecture, and training data. But we would like to clarify that **Vision Ratio is statistically associated with visual understanding performance, rather than to make a causal claim**.
> > >
> > > ### **1. Scope of our claim: statistical association rather than causality.**
> > > We would like to clarify that our claim about the relationship between Vision Ratio and visual understanding performance is intentionally correlational rather than causal. Therefore, we consistently describe Vision Ratio as **a useful indicator of downstream performance** (Abstract, Introduction) or report **a statistically positive association between Vision Ratio and visual understanding ability** (Section 4.3) in this paper.
> > >
> > > ### **2. Statistical evidence.**
> > > - In Section 4.3, we evaluate 10 diverse and widely used MLLMs and compare their Vision Ratio with the average accuracy over 7 widely used visual understanding benchmarks. We observe a statistically positive correlation, with **Spearman’s ρ = 0.964**.
> > >
> > > - Moreover, in our previous response **R3**, when we adjust the Vision Ratio within the same MLLM, the downstream task performance changes monotonically in the same direction across multiple tasks. This further supports that **Vision Ratio captures a behaviorally meaningful property that is statistically associated with downstream performance**.
> > >
> > > Overall, our results show that Vision Ratio, as a quantitative behavioral measure of modality preference, is statistically associated with downstream visual understanding performance and may serve as a useful behavioral signal.
> > >
> > > ### **3. Revisions for clearer scope and wording.**
> > >
> > > In the revised version, we will revise the wording in the abstract from "a useful indicator of downstream performance" to "statistically associated with downstream performance" to avoid unnecessary misunderstanding, and explicitly clarify that **our findings should be interpreted as a statistical association rather than a causal relationship**,
> > >
> > > Best regards,
> > >
> > > All Authors

---

### Official Review · Reviewer_6ENt · 2026-03-09

**Soundness:** 3
**Presentation:** 2
**Significance:** 2
**Originality:** 2
**Overall Recommendation:** 3
**Confidence:** 3

**Summary:**

This paper studies modality preference in multimodal large language models, asking whether they tend to rely more on images or text when the two modalities conflict. To measure this systematically, authors introduce MC2, a benchmark of 2,000 carefully constructed image–text conflict examples spanning 8 task categories, and define Vision Ratio as a metric for quantifying whether a model prefers visual or textual evidence. Their experiments on 20 MLLMs show that most models exhibit a strong text bias, while larger models generally become more vision-oriented, and this preference is highly correlated with downstream multimodal performance. Building on this finding, the paper further proposes a training-free representation engineering method that extracts a modality preference direction from hidden states and steers the model toward relying more on either vision or text at inference time, leading to improved performance on several downstream tasks.

**Compliance With Llm Reviewing Policy:**

Affirmed.

**Final Justification:**

The authors' response addresses most of my previous questions. However, I remain confused about the benchmark metric itself.

In the rebuttal, the authors argue that the interpretation of the “best” Vision Ratio depends on the task setting: for vision understanding tasks, a higher Vision Ratio appears to be better, while for text-dominant multimodal machine translation tasks, the desirable point seems closer to a more balanced setting (i.e., 0.5). I agree with this explanation.

However, this also highlights my main concern: the current benchmark seems to mainly evaluate vision understanding tasks, where the empirical trend is consistently that stronger models tend to have higher Vision Ratios. In this sense, the benchmark appears to reward a “higher is better” pattern. It does not seem to meaningfully evaluate text-dominant multimodal tasks, nor does it provide a clear mechanism for reflecting the authors’ claim that being closer to 0.5 is preferable when the goal is intrinsic modality-neutrality.

In other words, the benchmark currently evaluates primarily vision-dominant settings and lacks sufficient coverage of text-dominant tasks. As a result, it cannot adequately support the authors’ claim that 0.5 should be considered the optimal value. More importantly, if the notion of the “best” score is itself task-dependent, conceptually unclear, and not fully covered by the benchmark design, then it becomes difficult to understand how this benchmark can be reliably used to assess whether one model is actually better than another.

Therefore, although I appreciate the authors’ clarification, I am still inclined to maintain my current score, mainly because the benchmark metric remains conceptually ambiguous and insufficiently justified.

**Key Questions For Authors:**

Please see weaknesses.

**Limitations:**

yes

**Strengths And Weaknesses:**

Strengths

1. This paper proposes a benchmark to study which modality a model tends to prefer. This is a fundamental and important question.
2. This paper not only introduces a new benchmark, but also proposes a training-free method to steer the model’s modality preference.

Weaknesses

1. It remains unclear what range of Vision Ratio should be considered desirable in this benchmark. Should a higher Vision Ratio always be preferred, or is a value closer to 0.5 more ideal? The paper does not fully clarify what constitutes the most favorable modality preference.
2. In Tab1, I noticed that for models with relatively high Vision Ratio, such as Qwen2.5VL-7B and InternVL3, increasing attention to text leads to performance gains, whereas for other models, increasing attention to vision appears to be more beneficial. It would be helpful to also report the reverse-direction results. For example, what happens if Qwen2.5VL-7B is further steered toward vision instead?
3. It would be beneficial to include results on more recent strong multimodal baselines, such as Qwen3-VL. In addition, although this is not strictly necessary, I am also curious whether native multimodal models such as Qwen3.5 would exhibit modality preferences different from those of earlier foundational MLLMs.
4. Table 5 reports both benchmark performance and Vision Ratio across different models. However, it remains unclear whether performance can be systematically improved by steering models with Vision Ratio < 0.5 toward vision, and models with Vision Ratio > 0.5 toward text. A more complete ablation in both steering directions would make this conclusion more convincing.

---

> ### Author Rebuttal · Authors · 2026-03-31
>
> > W1: What is the desirable range of Vision Ratio?
>
> **R1:** Thanks for your question. We would like to discuss this from two distinct perspectives:
> 1. Ideally, information from different modalities is equally important for task solving, where a vision ratio of 0.5 ideal. This signifies that MLLMs **equally utilize contexts from both vision and text modalities**.
> 2. However, in the current benchmarks for real-world tasks, the evaluation itself exhibits a certain preference towards different modalities. Therefore, **aligning the preference of the guiding model towards the task's preference** can enhance task performances, which is empirically validated in Table 2 and Table 3
>
> > W2: Results for Steering Toward Inherent Preferences.
>
> **R2:** Thank you for your comment. Firstly, we would like to clarify that the reason for steering performance gains lies in **steering towards the modality required by the task**, rather than adjusting MLLMs' preferences in the opposite direction. In Table 1, we only aim to verify the method's **effectiveness in following the context in the opposite direction of the preference by steering towards the modality with the opposite preference**, achieving performance gains. To address your question, we steer MLLMs towards their inherent preferences to **evaluate the effect of making the model follow context in the opposite preference**. The results are as follows:
>
> | Model  | MLLM  | Ours |
> | :-| :-| :-|
> | Qwen2.5VL-7B  |  35.4 | 32.4|
> | InternVL3-8B  |  20.9 | 19.0|
> | Qwen2VL-7B  | 15.3 | 13.4|
> | OneVision-7B  | 37.5 | 33.6|
>
> The results indicate that steering MLLMs towards their inherent preferences can degrade the effect of following preference-contrary contexts, further validating the effectiveness of the proposed method. We will elaborate more clearly on these evaluations and clarifications in the revised version
>
>
> > W3: Add the results on Qwen3-VL and Qwen3.5-VL.
>
> **R3:** Thanks for your suggestion. we have added the results on Qwen3-VL and Qwen3.5-VL as follows:
>
> | Model | Vision Ratio  |
> | :- | :- |
> | Qwen2.5VL-7B  | 59.6 |
> | Qwen3-VL-8B  | 77.8 |
> | Qwen3-VL-30B-A3B  | 88.4 |
> | Qwen3.5-VL-9B  | 79.9 |
>
> We observe that both Qwen3-VL and Qwen3.5-VL exhibit a distinct vision preference. Furthermore, as the model size increases, the Qwen3-VL-30B-A3B demonstrates a more pronounced vision preference compared to the Qwen3-VL-8B, which aligns with the conclusions presented in Section 4.1.
>
> > W4: Comprehensive ablation in both steering directions in Table 5.
>
> **R4:** Firstly, we would like to clarify that **steering towards the modality required by the task can enhance performance**. However, the vision understanding tasks listed in Table 5 are all vision-centric tasks, so theoretically steering towards vision enhances performance, whereas steering towards text leads to a degradation. To address your question, we steer OneVision-7B and Qwen2.5VL-7B towards both text and vision to test the effects.
>
> The results are as follows:
>
> | Model |  MMMU | HallBench | RealworldQA |
> | :- | :- | :- | :- |
> | Qwen2.5VL-7B   | 58.6  | 52.9 | 68.5 |
> |+towards text   | 56.2  | 49.3 | 67.2 |
> |+towards vision | 58.9  | 53.4 | 70.5 |
> | OneVision-7B   | 47.9  | 31.6 | 66.3 |
> |+towards text   | 46.3  | 29.5 | 63.1 |
> |+towards vision | 48.1  | 33.2 | 68.2 |
>
> The results indicate that steering MLLMs toward vision enhances performance on these vision-centric tasks, whereas steering them toward text leads to a noticeable degradation. We will incorporate these clarifications in the revised version.

---

> > ### Author Rebuttal · Reviewer_6ENt · 2026-04-02
> >
> > Thank the authors for their response. However, I remain confused about the vision ratio. The authors claim that 0.5 is the ideal value. However, as model performance gradually improves, the vision ratio also continues to increase, which seems inconsistent with the authors’ explanation. Besides, this raises the question of whether the benchmark may be overly dependent on visual information, without maintaining a sufficient balance with the textual modality. In addition, the rebuttal results seem to suggest that steering the model toward the text modality consistently leads to worse performance.

---

> > > ### Author Response · Authors · 2026-04-03
> > >
> > > Thank you for your response! We appreciate the opportunity to further clarify the distinction between **intrinsic modality-neutrality** and **optimal task performance**. As discussed in our previous response **R1**, the "ideal" Vision Ratio should be interpreted from two distinct perspectives:
> > > 1. The desirable Vision Ratio for intrinsic modality-neutrality of MLLMs is 0.5.
> > > 2. The task's inherent dependency of evidence from different modalities dictates which Vision Ratio yields the best performance.
> > >
> > > Consequently, **Vision Ratio = 0.5 does not imply optimal performance across all downstream tasks**. We elaborate on these two perspectives below:
> > >
> > > 1. The desirable Vision Ratio for **intrinsic modality-neutrality of MLLMs** is 0.5. Vision Ratio = 0.5 signifies that MLLMs assign equal weight to visual and textual evidence, serving as a desirable intrinsic modality-neutral property of MLLMs. This is particularly desirable for building **more faithful**, **less-unbiased** AI assistants.
> > > For instance, in scenarios such as multimodal dialogue and RAG, this property ensures robust adherence to instructions under conflicting multimodal evidence, preventing the assistant from over-reliance on its modality prior.
> > >
> > > 2. The task's **inherent dependency of evidence from different modalities** dictates which Vision Ratio yields the best performance.
> > >
> > > - On the one hand, current vision understanding tasks require MLLMs to **treat text as the primary evidence source** as emphasized in [1,2], while treating text input as instruction. For these tasks, models more inclined to use visual evidence (higher vision ratio) will achieve stronger performance, as evidenced by the results in Fig. 3. Moreover, we also verify that steering towards vision can lead to better performances (**MathVista, TallyQA and VSR in Table 2 and PhD-icc, PhD-iac in Table 3**), whereas steering towards text leads to performance degradation in the previous response **R4**.
> > >
> > > - On the other hand, prior works [3,4,5] have identified text-dominant multimodal machine translation tasks, such as AmbigCaps, which requires models to **prioritize textual evidence** as the primary context while treating visual signals as auxiliary guidance. We investigate the correlation between performance (BLEU score) of AmbigCaps and the Vision Ratio, as summarized below:
> > > | Model | Vision Ratio | En->Tr | Tr->En |
> > > | :- | :- | :- | :- |
> > > | OneVision-7B | 26.3 | 9.56 | 19.51 |
> > > | InternVL3-9B | 41.2 | 9.32 | 19.10 |
> > > | InternVL3-14B | 55.0 | 8.90 | 18.89 |
> > > | Qwen2.5VL-7B | 59.6 | 8.92 | 18.56 |
> > >
> > > An **inverse relationship** exists between the Vision Ratio and AmbigCaps performance, indicating that **lower Vision Ratio benefits text-dominant multimodal tasks**. This is further supported by the AmbigCaps results in Table 2, where steering toward text yields performance gains on AmbigCaps, while steering toward vision results in performance degradation.
> > >
> > > Therefore, we view Vision Ratio = 0.5 as a desirable indicator of intrinsic modality-neutrality, while the best-performing Vision Ratio remains task-dependent.
> > >
> > > Finally, we acknowledge that our main analysis (Fig. 3, Table 2 and Table 3) is **based on vision understanding tasks except for AmbigCaps**, which may lead to a misunderstanding regarding the generalizability of our claims. To avoid this, we will **strictly clarify the scope of our conclusions** in the revision and explore a broader range of real-world applications where intrinsic modality-neutrality may matter in future work.
> > >
> > > We will explicitly clarify these in the revised version.
> > >
> > > [1] VLind-Bench: Measuring Language Priors in Large Vision-Language Models, NAACL 2025
> > >
> > > [2] Mitigating Object Hallucinations in Large Vision-Language Models through Visual Contrastive Decoding, CVPR 2024
> > >
> > > [3] Does Vision Still Help? Multimodal Translation with CLIP-Based Image Selection, ACL 2025
> > >
> > > [4] Imagination and Contemplation: A Balanced Framework for Semantic-Augmented Multimodal Machine Translation, EMNLP 2025
> > >
> > > [5] Multimodal Machine Translation with Text-Image In-depth Questioning, ACL 2025
> > >
> > >
> > > Best regards,
> > >
> > > All authors

---

### Official Review · Reviewer_gAqo · 2026-03-11

**Soundness:** 3
**Presentation:** 3
**Significance:** 3
**Originality:** 3
**Overall Recommendation:** 4
**Confidence:** 3

**Summary:**

This paper systematically investigates modality preferences in Multi-modal Large Language Models (MLLMs). To eliminate confounding factors, the authors introduce the $MC^2$ benchmark, comprising 2,000 carefully curated image-text conflict samples across diverse tasks. Evaluating 20 MLLMs reveals a prevalent text preference, while visual preference strongly correlates with downstream task performance. Furthermore, the authors demonstrate that modality preference is identifiable in the latent space. Consequently, they propose a training-free representation engineering method to explicitly probe and steer this preference, yielding significant improvements in multi-modal understanding and reasoning tasks.

**Compliance With Llm Reviewing Policy:**

Affirmed.

**Final Justification:**

Most of My concerns have been addressed after the authors' rebuttal. The author also commit to release all code and data. Thus, I maintain my positive score.

**Key Questions For Authors:**

Please refer to the 'Weaknesses' section above.

**Limitations:**

yes

**Strengths And Weaknesses:**

**Strengths**
1. The paper introduces a highly original approach to evaluating modality preference by constructing controlled evidence-conflict scenarios rather than relying on the biased method of simply masking inputs.
2. The methodology is technically rigorous, utilizing a robust semi-automated pipeline that incorporates strict human verification to construct the $MC^2$ benchmark. Furthermore, the authors perform comprehensive ablation studies and PCA to validate the stability and effectiveness of their representation engineering approach across various layers and steering intensities.
3. The manuscript is exceptionally well-structured, clearly articulating its motivation, data construction philosophy, and empirical findings in a logical flow.

**Weaknesses**
1. The proposed modality preference steering technique lacks algorithmic novelty, as it essentially directly applies the "In-context vectors" (ICV) approach introduced by Liu et al. (2023)  to the multi-modal domain. While utilizing this for modality preference is an interesting application, the core mathematical formulation—subtracting hidden states of contrasting inputs and adding the scaled vector during inference —is heavily borrowed without significant architectural adaptation for MLLMs.

2. The experimental validation relies heavily on the rigid binary-choice format of the constructed $MC^2$ benchmark. The paper lacks comprehensive evaluations on open-ended visual question answering or long-form generation tasks, leaving it unclear whether applying this global steering vector inadvertently degrades the model's free-form text generation coherence or fluency.

3. Although the paper provides PCA visualizations and attention score plots, the analysis remains largely empirical rather than mechanistic. For instance, the authors empirically identify that middle-to-late layers (e.g., layers 20-23) exhibit higher absolute values for preference direction, but they fail to provide a rigorous theoretical or mechanistic explanation for why these specific layers capture modality preference, weakening the depth of the overall findings.

Reference:

Liu, S., Ye, H., Xing, L., & Zou, J. (2023). In-context vectors: Making in context learning more effective and controllable through latent space steering. arXiv preprint arXiv:2311.06668.

---

> ### Author Rebuttal · Authors · 2026-03-31
>
> Thanks for your valuable comments! We clarify these questions point by point and include additional details in the anonymous link (https://anonymous.4open.science/r/Modality-Preference-8016).
>
> > W1: Algorithmic advances and architectural adaptation for MLLMs.
>
> **R1:** Thanks for your comment. The proposed method mainly features two critical improvements compared to ICV for MLLMs:
>
> 1) Unlike **token-wise steering** in ICV, we apply a unified steering vector (**identical strength and direction**) to both vision and text tokens. As noted by [2], vision and text representations in MLLMs are often out-of-distribution relative to each other. Consequently, applying disparate steering vectors maybe lead to a **significant distortion of the relative distribution** between vision and text modalities, thereby undermining the model's intrinsic capabilities. Our experiments on Qwen2-VL-7B confirm that using ICV's token-wise steering leads to significant performance degradation:
>
> |Method|MC2|Phd-icc|Phd-iac|
> | :---:|:---:|:---:|:---:|
> |Qwen2-VL-7B|15.3|6.0|27.6|
> |ICV|38.9|13.6|30.1|
> |Ours|48.1|18.4|36.1|
>
> 2) The proposed method can **adaptively** adjust the steering weight without the need for manual enumeration of steering weights like ICV.
>
> We will include these clarifications in the revised version to better highlight the algorithmic improvements upon ICV.
>
> > W2: Validation on open-ended or long-form generation tasks.
>
> **R2:** Thanks for your comment. Though the constructed benchmark is binary-choice, we have conducted comprehensive experiments on open-ended VQA tasks (VSR, TallyQA) and long-form generation task (Ambigcaps-Translation) in Table 2, summarized as follows:
>
> |           Dataset           |    CoT     | InstDesign |    Ours    |
> | :-------------------------: | :--------: | :--------: | :--------: |
> |           TallyQA           |    61.6    |    74.4    |    76.6    |
> |             VSR             |    45.6    |    51.3    |    53.2    |
> | Ambigcaps (En->Tr & Tr->En) | 9.03/18.63 | 9.45/18.98 | 10.2/19.89 |
>
> These results show that the proposed method improves performance on both open-ended VQA and long-form generation tasks.
>
> Besides, according to your suggestion, we evaluate the **generation coherence** (measured by PPL) and **fluency** (measured by GPT-win rate) on the free-form text generation tasks (**Ambigcaps** with 1000+ samples), results (en->tr/tr->en) are as follows:
>
> |Method|PPL(Lower is better)|GPT-win rate(Higher is better)|
> |:-----:|:---:|:---:|
> |MLLM|33.8/54.4|45.8/53.4|
> |Ours|24.2/60.8|54.2/46.6|
>
> As shown, our method maintains competitive generation quality without catastrophic degradation.
>
> > W3: Mechanistic explanation of the identified steering layers.
>
> **R3:** First, we want to emphasize that the identification method of steering layers can be **generalized to multiple MLLMs (LLaVA1.5-7B, InternVL3, OneVision, Qwen2.5VL, etc.) and downstream tasks**, which demonstrates the robustness of the practical applications.
>
> Second, we thank the reviewer for the insightful suggestions to delve deeper into **the mechanistic underpinnings of our findings**. To provide a more rigorous explanation, we track the evolution of modality preference within the residual stream through a **mechanistic logit lens method** [3]. Specifically, we project the layer-wise hidden states of the last token into Unembedding matrix of Qwen2.5VL-7B under modal conflict prompts. And we calculate the **Modality Preference Margin** (MPM) - the average logit difference between the preferred and competitor modalities.
> The results (**Text_Preference.png and** and **Vision_Preference.png** in the anonymous link) are summarized below:
>
> |Layer|0-19|20|21|22|23|24|25|26|27
> |:-----:|:---:|:---:|:---:|:---:|:---:|:---:|:---:|:---:|:---:|
> |Vision Preference Margin|Nearly 0|0.11|0.15|0.34|0.78|1.58|2.44|3.79|4.92|
> |Text Preference Margin|Nearly 0|0.05|0.11|0.17|0.45|0.93|1.30|1.94|2.95|
>
> The MPM remains near zero for layers 0-19, indicating that early layers focus on raw feature extraction. Crucially, the preference margin first surfaces at **Layers 20-23**, marking it as the critical decision point for preference arbitration. While later layers (24+) have larger margins, they represent the decision already made. Therefore layers **20-23 constitute the most effective intervention point for steering modality preference**, which aligns with our empirical findings. We will incorporate these mechanistic results into the revised version.
>
> [1] In-context vectors: Making in context learning more effective and controllable through latent space steering
>
> [2] Unveiling Intrinsic Text Bias in Multimodal Large Language Models through Attention Key-Space Analysis
>
> [3] Transformer feed-forward layers build predictions by promoting concepts in the vocabulary space, EMNLP 2022

---

> > ### Author Rebuttal · Reviewer_gAqo · 2026-04-02
> >
> > Thank you for the response. Given the explanation and new evidence, I maintain my score.

---

> > > ### Author Response · Authors · 2026-04-03
> > >
> > > Dear Reviewer gAqo,
> > >
> > > Thank you very much for your timely response. We are pleased to note that **our explanations and the new evidence provided during the rebuttal have successfully addressed your all concerns**.
> > >
> > > We sincerely appreciate your generous recognition of our work, especially regarding our following strengths:
> > > 1. **A highly original and rigorous benchmark**: The paper evaluates modality preference in MLLMs by eliminating confounding factors rather than relying on the biased method of simply masking inputs.
> > > 2. **Comprehensive experiment analysis**: The paper conducts comprehensive ablation studies and PCA to validate the stability and effectiveness of the proposed approach.
> > > 3. **Exceptionally well-structured manuscript**: This paper clearly articulate its motivation, data construction philosophy, and empirical findings in a logical flow.
> > >
> > > We are particularly grateful for your insightful suggestion to **investigate the mechanistic underpinnings of our findings**. The mechanistic logit lens analysis has provided a deeper perspective on how preference arbitration occurs within the model's layers.
> > >
> > > We committed to incorporating these analysis experiments and mechanistic insights into the final version. Furthermore, we will release all code and data to ensure full reproducibility and to support continued research within the multi-modal community.
> > >
> > > Thank you again for your constructive review and for supporting our work.
> > >
> > > Best regards,
> > >
> > > All Authors

---

### Official Review · Reviewer_STB3 · 2026-03-14

**Soundness:** 3
**Presentation:** 2
**Significance:** 3
**Originality:** 3
**Overall Recommendation:** 4
**Confidence:** 3

**Summary:**

This paper explores the modality preference in VLMs, which is an important problem in multi-modal model training. They build a test set to dialog some vlms and have some findings.

**Compliance With Llm Reviewing Policy:**

Affirmed.

**Final Justification:**

The authors addressed most of my concerns during the rebuttal, which significantly improved the clarity of the work. Therefore, I have increased my score to weak accept.

**Key Questions For Authors:**

Refer to above.

**Limitations:**

It can make the paper’s main messages easier to grasp and provide readers with genuinely actionable findings.

**Strengths And Weaknesses:**

Strengths
1. The overall motivation is sound — the imbalance among modalities is indeed a key issue in multimodal training.
2. The paper includes a substantial number of experiments that are straightforward and easy to understand.

Weaknesses
1. Most MLLMs (except Qwen2.5-VL and InternVL3) display text preference. Why does Qwen2.5-VL behave differently?
2. Could the modality preference be related to the token length or composition? Qwen2.5-VL has a visual token ratio of 59.6% — what does this imply? Does a higher visual ratio necessarily lead to better performance?
3. The paper states that larger MLLMs exhibit stronger visual preferences — why would model size lead to such a trend?
4. The phrase “Vision Ratio aligns with human preference” is unclear. Does this mean human preference follows the same pattern as large model behavior?
5. You claim that “Qwen2.5VL-7B exhibits greater shifts than Qwen2.5VL-3B under the same training data, supporting Hypothesis 2.” To make this conclusion convincing, it would be advisable to include comparisons with either smaller or larger models.
6. The figures (e.g., Figure 2 and Figure 3) could be better aligned with the conclusions. Currently, readers need to flip back and forth to connect figures with corresponding analyses.
7. The statement “Modality preference can be guided through instruction design” suggests that prompt modification can influence modality preference — is this interpretation correct? Please clarify.
8. The overall conclusion of the paper is not sufficiently highlighted and would benefit from revision to make the key takeaways more prominent.
9. The overall contribution still feels somewhat limited and could be further emphasized to better highlight your main advances.

---

> ### Author Rebuttal · Authors · 2026-03-31
>
> Thanks for valuable comments! We clarify each point below, include additional details in anonymous https://anonymous.4open.science/r/Modality-Preference-8016 and commit to including all new experiments and clarifications in the revised version.
> >W1 & W3: Origins of Modality Preference
>
> **R1 & R3:** MLLMs are typically initialized from **LLMs with inherent text preference**, and then adopt vision-centric training tasks to acquire vision understanding ability. As discussed in Section 4.2, we find that the divergence in modality preference is driven by training factors:
>
> 1. Qwen2.5-VL's training frequently contains **integrated vision-and-text contexts**, that trigger **cross-modal contention**, forcing the model to resolve inconsistencies by prioritizing visual cues. In contrast, MLLM like LLaVA-1.5 is trained on vision-as-context tasks, thus retaining more of their initial text preference.
> 2. Larger MLLMs exhibit **superior adaptability in shifting modality preferences**. When optimized for vision understanding, larger models are more likely to achieve a more pronounced vision preference.
>
> These are validated **in Fig 3** and the experiments of larger model in **R5**.
> >W2: Impact of Token Length on Vision Ratio & Practical Meaning of Vision Ratio
>
> **R2-1:** To investigate the impact of token length on modality preference, we adjusted the text length to approximate 0.6x–1.3x of the original, while ensuring the core semantics remain intact, to observe changes in Vision Ratio, as follows:
>
> |Length|0.6|0.8|1.0|1.2|1.3|
> |:-|:-|:-|:-|:-|:-|
> |Qwen2.5VL-7B|60.4|60.2|59.6|59.4|59.3|
> |InternVL3-14B|56.9|54.6|55.0|53.8|54.2|
> |OneVision-7B|25.7|25.5|26.3|25.2|25.3|
>
> Results show that varying the textual length does not induce any significant shifts in modality preference.
>
> **R2-2:** Statistically, Qwen2.5-VL-7B favors visual evidence in 59.6% of conflict scenarios. A higher vision ratio can enhance the performance of vision-dominated tasks, whereas a lower ratio can favor the performance of text grounding tasks. As shown in Table 3 and Table 4, steering towards vision improves performance in visual grounding tasks, while steering towards text benefits tasks requiring precise textual grounding, such as multimodal machine translation
>
> >W4: Clarify "Vision Ratio aligns with human preference"
>
> **R4:** We apologize for the confusion. We clarify that Vision Ratio serves as a reliable automated proxy for human assessments of an MLLM's modality preference. We will revise it to avoid any potential confusion in the revised version.
> >W5: More models to support Hypothesis 2
>
> **R5:** Following your suggestion, we supplemented our analysis with the Qwen2.5-VL-32B model to further validate Hypothesis 2 (as no <3B model is currently available in this family), as follows:
>
> |Proportion of Conflict Modal|original|0.0|0.25|0.5|0.75|1.0|
> |:-|:-|:-|:-|:-|:-|:-|
> |Qwen2.5VL-3B|33.6|32.4|37.8|37.3|37.9|36.6|
> |7B|51.9|50.7|52.5|52.8|55.6|57.3|
> |32B|67.9|67.8|72.4|74.2|78.0|83.2|
>
> We observe that the 32B model exhibits a sharper Vision Ratio shift than 7B and 3B variants, as the training data of conflict modal context increases.
> >W6: Better align the figures with conclusions
>
> **R6:** Thanks for your suggestion! We revised the figures by adding concise analytical summaries into the captions. The updated versions (Fig2-6.png) are available in the anonymous link.
> >W7: Clarify "Modality preference can be guided through instruction design"
>
> **R7:** Modality preference is an intrinsic and foundational behavioral tendency inherent to MLLMs. What we intend to convey is that instructions can modulate the **manifested expression of modality preference**, rather than the inherent modality preference itself. To avoid ambiguity, we will revise it as "The manifested output expression of modality preference can be guided through instruction design".
> >W8 & W9 & Limitation: Highlight overall conclusion and contribution and provide actionable insights
>
> **R8:** Thanks for constuctive suggestions! We will better highlight our Key Takeaways, primary advances and actionable insights as follows:
>
> **Key Takeaways**
> 1. Modality preference is ubiquitous across MLLMs and is a critical indicator of MLLM capability.
> 2. Modality preference is driven by training factors, warranting more attention from large-scale training.
> 3. Latent steering proves that modality preference is both controllable and highly correlated with task performance.
>
> **Primary Advances**
> 1. Rigorous Evaluation: We rigorously assess modality preference by decoupling it from confounding variables.
> 2. New Findings: Beyond the assessment of preference, we demonstrate latent controllability and establish its causal link to downstream performance.
>
> **Actionable Insights**
> 1. Vision Ratio serves as an efficient metric for assessing a model's visual understanding.
> 2. In omni-modal contexts, modality preference acts as a key measure for tracking capability retention and modality adaptation.

---

> > ### Author Rebuttal · Reviewer_STB3 · 2026-04-04
> >
> > Thank you for the additional ablation studies and the detailed clarifications. I appreciate the authors’ effort in addressing my questions and strengthening the empirical support of the paper. While the current summary points are still not sufficiently detailed or explicit, they have addressed most of my questions. Based on this, I am willing to increase my score.

---

> > > ### Author Response · Authors · 2026-04-04
> > >
> > > Dear Reviewer STB3,
> > >
> > > Thanks for your thoughtful and encouraging response, and we are grateful that **you are willing to increase your score**!
> > >
> > > We also appreciate your suggestion that our current summary points should be made more detailed and explicit. In response to this suggestion, we provide a more explicit and detailed version of our **Key Takeaways**, **Primary Advances**, and **Actionable Insights** as follows:
> > >
> > > ## **Key Takeaways:**
> > >
> > > 1. Most models exhibit a text preference. Some models, such as Qwen2.5VL and InternVL3, display a visual preference, and larger models tend to show stronger vision preference.
> > >
> > > 2. Modality preference is shaped by training factors, including the training recipe and model scale. Given its positive statistically associated relationship with downstream task performance, Vision Ratio may serve as a rough proxy for assessing a model’s visual understanding.
> > >
> > > 3. Modality preference is controllable in latent space. By steering modality preference through hidden-state intervention, we can not only adjust the inherent modality preference of MLLMs, but also improve downstream task performance.
> > >
> > > ## **Primary Advances:**
> > >
> > > 1. Unlike prior studies based on isolated unimodal inputs, we evaluate modality preference in scenarios where visual and textual evidence co-exist, and explicitly decouple it from confounding factors to rigorously assess the modality preference.
> > >
> > > 2. We find that modality preference is widespread across different architectures and sizes, and that its origins are shaped by training factors such as model scale and the training recipe.
> > >
> > > 3. We demonstrate that modality preference is controllable in latent space and further show that adjusting it can improve downstream task performance.
> > >
> > > ## **Actionable Insights:**
> > >
> > > 1. Vision Ratio may serve as a lightweight proxy for the assessment of a model's visual understanding ability. This makes it a potentially useful and efficient diagnostic signal during large-scale training and evaluation.
> > >
> > > 2. In omni-modal training and continual adaptation scenarios, modality preference may provide a useful indicator of how well the model retains its capabilities in original modalities while learning or optimizing for a newly introduced modality.
> > >
> > > 3. The presence of modality preference suggests that MLLMs may develop imbalanced learning behaviors during multimodal training. Consequently, it may be helpful to consider balanced multimodal optimization to support capability improvements in future omni-modal models.
> > >
> > > We will incorporate a brief summary of these key takeaways at the beginning of the corresponding subsections in **Section 4** and **Section 5**. And we will highlight these primary advances in the **Abstract** and **Introduction**, and summarize the actionable insights in the **Discussion** and **Conclusion**.
> > >
> > > Additionally, **we noticed that the score has not yet been updated, and we completely understand that there may be system delays or other factors at play. Please let us know if there is any additional information we could provide to support your assessment.**
> > >
> > > Thank you again for your constructive review and for supporting our work!
> > >
> > > Best regards,
> > >
> > > All Authors

---

### Decision · Program_Chairs · 2026-04-30

**Decision:**

Accept (regular)

**Comment:**

Reviewers acknowledge the originality of this paper, consistently find the problem of modality imbalance/preference in multimodal learning practical and important, and praise the extensive empirical results. In the period of rebuttal, all reviewers acknowledge the authors' effort in addressing their concerns, resulting in three out of four reviewers favoring weak accept.
Clearly there are still remaining concerns, but most reviewers find merits of this paper outweigh weaknesses. So I recommend weak accept, and urge authors to incorporate all reviewers' suggestions in their revision, especially the one regarding the ideal value choice of vision ratio.